# Containing a spread through sequential learning: to exploit or to explore?

**Xingran Chen**                                    *xingranc@seas.upenn.edu*
*Electrical and Systems Engineering Department*
*University of Pennsylvania*

**Hesam Nikpey**                                    *hesam@seas.upenn.edu*
*Computer and Information Science Department*
*University of Pennsylvania*

**Jungyeol Kim**                                    *jungyeol@alumni.upenn.edu*
*JPMorgan Chase & Co.*

**Saswati Sarkar**                                    *swati@seas.upenn.edu*
*Electrical and Systems Engineering Department*
*University of Pennsylvania*

**Shirin Saeedi-Bidokhti**                                    *saeedi@seas.upenn.edu*
*Computer and Information Science Department*
*University of Pennsylvania*

**Reviewed on OpenReview:** *https://openreview.net/pdf?id=qvRWcDXBam*

## Abstract

The spread of an undesirable contact process, such as an infectious disease (e.g. COVID-19), is contained through testing and isolation of infected nodes. The temporal and spatial evolution of the process (along with containment through isolation) render such detection as fundamentally different from active search detection strategies. In this work, through an active learning approach, we design testing and isolation strategies to contain the spread and minimize the cumulative infections under a given test budget. We prove that the objective can be optimized, with performance guarantees, by greedily selecting the nodes to test. We further design reward-based methodologies that effectively minimize an upper bound on the cumulative infections and are computationally more tractable in large networks. These policies, however, need knowledge about the nodes' infection probabilities which are dynamically changing and have to be learned by sequential testing. We develop a message-passing framework for this purpose and, building on that, show novel tradeoffs between exploitation of knowledge through reward-based heuristics and exploration of the unknown through a carefully designed probabilistic testing. The tradeoffs are fundamentally distinct from the classical counterparts under active search or multi-armed bandit problems (MABs). We provably show the necessity of exploration in a stylized network and show through simulations that exploration can outperform exploitation in various synthetic and real-data networks depending on the parameters of the network and the spread.

## 1 Introduction

We consider learning and decision making in networked systems for processes that evolve both temporally and spatially. An important example in this class of processes is COVID-19 infection. It evolves in time (e.g. through different stages of the disease for an infected individual) and over a contact network and its spread can be contained by testing and isolation. Public health systems need to judiciously decide who should be

tested and isolated in presence of limitations on the number of individuals who can be tested and isolated on a given day.

Most existing works on this topic have investigated the spread of COVID-19 through dynamic systems such SIR models and their variants [1, 2, 3, 4, 5, 6]. These models are made more complex to fit the real data in [7, 8, 9, 10, 11, 12]. Estimation of the model parameters by learning-based methods are considered and verified by real data in [13, 14, 15, 16, 17, 18]. Other attributes such as lockdown policy [19], multi-wave prediction [20], herd immunity threshold [21] are also considered by data-driven experiments. These works mostly focus on the estimation of model parameters thorough real data, and aim to make a more accurate prediction of the spread. None of them, however, consider testing and isolation policies. Our work complements these investigations by designing sequential testing and isolation policies in order to minimize the cumulative infections. For this purpose, we have assumed full statistical knowledge of the spread model and the underlying contact network and we are not concerned with prediction and estimation of model parameters.

Designing optimal testing and control policies in dynamic networked systems often involves computational challenges. These challenges have been alleviated in control literature by capturing the spread through differential equations [22, 23, 24, 25, 26]. The differential equations rely on classical mean-field approximations, considering neighbors of each node as "socially averaged hypothetical neighbors". Refinements of the mean-field approximations such as pair approximation [27], degree-based approximation [28], meta-population approximation [29] etc, all resort to some form of averaging of neighborhoods or more generally groups of nodes. The averaging does not capture the heterogeneity of a real-world complex social network and in effect disregards the contact network topology. But, in practice, the contact network topologies are often partially known, for example, from contact tracing apps that individuals launch on their phones. Thus testing and control strategies must exploit the partial topological information to control the spread. The most widely deployed testing and control policy, the (forward and backward) contact tracing (and its variants) [30, 31, 32, 33, 34, 35, 36, 37, 38], relies on partial knowledge of the network topology (ie, the neighbors of infectious nodes who have been detected), and therefore does not lend itself to mean-field analysis. Our proposed framework considers both the SIR evolution of the disease for each node and the spread of the disease through a given network.

The following challenges arise in the design of intelligent testing strategies if one seeks to exploit the spatio-temporal evolution of the disease process and comply with limited testing budget. Observing the state of a node at time $t$ will provide information about the state of (i) the node in time $t+1$ and (ii) the neighbors of the node at time $t, t+1, \ldots$. This is due to the inherent correlation that exists between states of neighboring nodes because an infectious disease spreads through contact. Thus, testing has a dual role. It has to both detect/isolate infected nodes and learn the spread in various localities of the network. The spread can often be silent: an undetected node (that may not be particularly likely to be infected based on previous observations) can infect its neighbors. Thus, testing nodes that do not necessarily appear to be infected may lead to timely discovery of even larger clusters of infected nodes waiting to explode. In other words, there is *an intrinsic tradeoff between exploitation of knowledge vs. exploration of the unknown.* Exploration vs. exploitation tradeoffs were originally studied in classical multi-armed bandit (MAB) problems where there is the notion of a single optimal arm that can be found by repeating a set of fixed actions [39, 40, 41]. MAB testing strategies have also been designed for exploring partially observable networks [42]. Our problem differs from what is mainly studied in the MAB literature because (i) the number of arms (potential infected nodes) is time-variant and actions cannot be repeated; (ii) the exploration vs. exploitation tradeoff in our context arises due to lack of knowledge about the time-evolving set of infected nodes, rather than lack of knowledge about the network or the process model and its parameters.

Note that contact tracing policies are in a sense exploitation policies: upon finding positive nodes, they exploit that knowledge and trace the contacts. While relatively practical, they have two main shortcomings, as implemented today: (i) They are not able to prioritize nodes based on their likelihood of being infected (beyond the coarse notion of contact or lack thereof). For example, consider an infectious node that has two neighbors, with different degrees. Under current contact tracing strategies, both neighbors have the same status. But in order to contain the spread as soon as possible, the node with a large degree should be prioritized for testing. A similar drawback becomes apparent if the neighbors themselves have a different

number of infectious neighbors; one with a larger number of infectious neighbors should be prioritized for testing, but current contact tracing strategies accord both the same priority. (ii) Contact tracing strategies do not incorporate any type of exploration. This may be a fundamental limitation of contact tracing. [38] has shown that, with high cost, contact tracing policies perform better when they incorporate exploration (active case finding). In contrast, our work provides a probabilistic framework to not only allow for exploitation in a fundamental manner but also to incorporate exploration in order to minimize the number of infections.

Finally, our problem is also related to active search in graphs where the goal is to test/search for a set of (fixed) target nodes under a set of given (static) similarity values between pairs of nodes [43, 44, 45, 46]. But the target nodes in these works are assumed fixed, whereas the target is dynamic in our setting because the infection spreads over time and space (i.e, over the contact network). Thus, a node may need to be tested multiple times. The importance of exploitation/exploration is also known, implicitly and/or explicitly, in various reinforcement learning literature [47, 48, 49].

We now distinguish our work from testing strategies that combine exploitation and exploration in some form [50, 51, 52]. Through a theoretical approach, [50] models the testing problem as a partially observable Markov decision process (POMDP). An optimal policy can, in principle, be formulated through POMDP, but such strategies are intractable in their general form (and heuristics are often far from optimal) [53, 54]. [50] devises tractable approximate algorithms with a significant caveat: In the design, analysis, and evaluation of the proposed algorithms, it is assumed that at each time the process can spread only on a single random edge of the network. This is a very special case that is hard to justify in practice and it is not clear how one could go beyond this assumption. On the other hand, [51] proposes a heuristic by implementing classical learning methods such as Linear support vector machine (SVM) and Polynomial SVM to rank nodes based on a notion of risk score (constructed by real-data) while reserving a portion of the test budget for random testing which can be understood as exploration. No spread model or contact network is assumed. [52] and this work were done concurrently. In [52], a tractable scheme to control dynamical processes on temporal graphs was proposed, through a POMDP solution with a combination of Graph Neural Networks (GNN) and Reinforcement Learning (RL) algorithm. Nodes are tested based on some scores obtained by the sequential learning framework, but no fundamental probabilities of the states of nodes were revealed. Different from [51, 52], our approach is model-based and we observe novel exploration-exploitation tradeoffs that arise not due to a lack of knowledge about the model or network, but rather because the set of infected nodes is unknown and evolves with time. We can also utilize knowledge about both the model and the contact network to devise a probabilistic framework for decision making.

We now summarize the contribution of some significant works that consider only exploitation and do not utilize any exploration [36, 37, 38]. [36, 37] have considered a combination of isolation and contact tracing sequential policies, and [36] has shown that the sequential strategies would reduce transmission more than mass testing or self-isolation alone, while [37] has shown that the sequential strategies can reduce the amount of quarantine time served and cost, hence individuals may increase participation in contact tracing, enabling less frequent and severe lockdown measures. [38] have proposed a novel approach to modeling multi-round network-based screening/contact tracing under uncertainty.

**Our Contributions**   In this work, we study a spread process such as Covid-19 and design sequential testing and isolation policies to contain the spread. Our contributions are as follows.

- Formulating the spread process through a compartmental model and a given contact network, we show that the problem of minimizing the total cumulative infections under a given test budget reduces to minimizing a supermodular function expressed in terms of nodes' probabilities of infection and it thus admits a *near-optimal greedy* policy. We further design reward-based algorithms that minimize an upper bound on the cumulative infections and are computationally more tractable in large networks.

- The greedy policy and its reward-based derivatives are applicable if nodes' probabilities of infection were known. However, since the set of infected nodes are unknown, these probabilities are unknown and can only be learned through *sequential testing*. We provide a message-passing framework for sequential estimation of nodes' *posterior* probabilities of infection given the history of test observations.

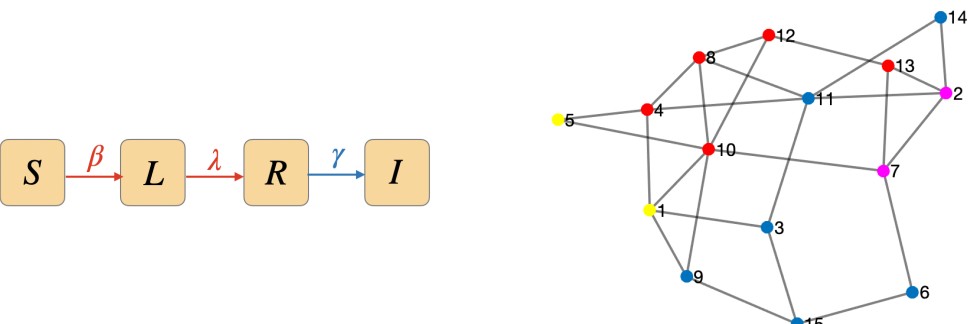

Figure 1: Left: Time evolution of the process per individual nodes. Right: A contact network with nodes in states susceptible (blue), latency (pink), infectious (red), recovered (yellow).

- We argue that testing has a dual role: (i) discovering and isolating the infected nodes to contain the spread, and (ii) providing more accurate estimates for nodes' infection probabilities which are used for decision making. In this sense, *exploitation* policies in which decision making only targets (i) can be suboptimal. We prove in a stylized network that when the belief about the probabilities is wrong, exploitation can be arbitrarily bad, while a policy that combines exploitation with random testing can contain the spread. This points to novel *exploitation-exploration tradeoffs* that stem from the lack of knowledge about the location of infected nodes, rather than the network or spread process.

- Following these findings, we propose *exploration* policies that test each node probabilistically according to its reward. The core idea is to balance exploitation of knowledge (about the nodes' infection probabilities and the resulting rewards) and exploration of the unknown (to get more accurate estimates of the infection probabilities). Through simulations, we compare the performance of exploration and exploitation policies in several synthetic and real-data networks. In particular, we investigate the role of three parameters on when exploration outperforms exploitation: (i) the *unregulated delay*, i.e., the time period when the disease spreads without intervention; (ii) the *global clustering coefficient* of the network, and (iii) the average *shortest path length* of the network. We show that when the above parameters increase, exploration becomes more beneficial as it provides better estimates of the nodes' probabilities of infection.

## 2 Modeling

To describe a spread process, we use a *discrete time compartmental model* [55]. Over decades, compartmental models have been key in the study of epidemics and opinion dynamics, albeit often disregarding the network topology. In this work, we capture the spread on a given contact network. For clarity of presentation, we focus on a model for the spread of COVID-19. The ideas can naturally be generalized to other applications. The main notations in the full paper are given in Table 1.

We model the progression of Covid-19 per individual, in time, through four stages or states: *Susceptible (S)*, *Latent (L)*, *Infectious (I)*, and *Recovered (R)*. Per contact, an infectious individual infects a susceptible individual with transmission probability $\beta$. An infected individual is initially in the latent state $L$, subsequently he becomes infectious (state $I$), finally he recovers (state $R$). Fig. 1 (left) depicts the evolution. The durations in the latent and infectious states are geometrically distributed, with means $1/\lambda, 1/\gamma$ respectively. We represent the state of node $i$ at time $t$ by random variable $\sigma_i(t)$ and its support set $\mathcal{X} = \{S, L, I, R\}$. We assume that the parameters $\beta$, $\lambda$ and $\gamma$ are known to the public health authority. This is a practical assumption because the parameters can be estimated by the public health authority based on the pandenmic data collected [56, 57, 58].

Let $\mathcal{G}(t) = (\mathcal{V}(t), \mathcal{E}(t))$ denote the contact network at time $t$, where $\mathcal{V}(t)$ is the set of nodes/individuals, of cardinality $N(t)$, and $\mathcal{E}(t)$ is the set of edges between the nodes, describing interactions/contacts on day $t$. Let $\mathcal{V} = \mathcal{V}(0)$, $\mathcal{E} = \mathcal{E}(0)$, $\mathcal{G} = \mathcal{G}(0)$, and $N = N(0)$. The network is time-dependent not only because

| Notations | Definitions |
|---|---|
| $\beta$ | transmission probability |
| $1/\gamma$ | mean duration in the latent state |
| $1/\lambda$ | mean duration in the infectious state |
| $\sigma_i(t)$ | state of node $i$ at time $t$, $\sigma_i(t) \in \{I, S, L, R\}$ |
| $\mathcal{G}(t)$ | contact network at time $t$ |
| $\mathcal{V}(t)$ | set of nodes at time $t$ |
| $\mathcal{E}(t)$ | set of edges at time $t$ |
| $N(t)$ | cardinality of $\mathcal{V}(t)$ |
| $N$ | $N = N(0)$ |
| $\partial_i(t)$ | neighbors of node $i$ at time $t$ |
| $\partial_i^+(t)$ | $\{i\} \cup \partial_i(t)$ |
| $Y_i(t)$ | testing result of node $i$ at time $t$ |
| $\mathcal{O}(t)$ | set of nodes tested at time $t$ |
| $\underline{Y}(t)$ | $\{Y_i(t)\}_{\{i \in \mathcal{O}(t)\}}$ |
| $B(t)$ | testing budget at time $t$ |
| $\pi$ | a testing and isolation policy |
| $C^\pi(t)$ | cumulative infections at time $t$ |
| $\mathcal{K}^\pi(t)$ | set of nodes tested at time $t$ (under policy $\pi$) |
| $K^\pi(t)$ | $K^\pi(t) = |\mathcal{K}^\pi(t)|$ |
| $T$ | time horizon |
| $\underline{v}_i(t)$ | true probability vector of node $i$ |
| $\underline{u}_i(t)$ | prior probability vector of node $i$ |
| $\underline{w}_i(t)$ | posterior probability vector of node $i$ |
| $\underline{e}_i(t)$ | updated posterior probability vector of node $i$ |
| $r_i(t)$ | rewards of selecting node $i$ at time $t$ |
| $\hat{r}_i(t)$ | estimated rewards of node $i$ at time $t$ |
| $\Psi_i(t)$ | $\Psi_i(t) = \mathcal{O}(t) \cap \partial_i^+(t-1)$ |
| $\Phi_i(t)$ | $\Phi_i(t) = \{j | j \in \partial_k^+(t-1), k \in \Psi_i(t)\} \backslash \{i\}$ |
| $\theta_i(t)$ | $\theta_i(t) = \sigma_i(t)|_{\{\underline{Y}(\tau)\}_{\tau=1}^{t-1}}$, $\theta_i(t) \in \{I, S, L, R\}$ |
| $\zeta_i(t)$ | $\zeta_i(t) = \sigma_i(t)|_{\{\underline{Y}(\tau)\}_{\tau=1}^{t}}$, $\zeta_i(t) \in \{I, S, L, R\}$ |

Table 1: Summary of main notations

interactions change on a daily basis, but also because nodes may be tested and isolated. If a node is tested positive on any day $t$, it will be isolated immediately. If a node is isolated on any day $t$, we assume that it remains in isolation until he recovers. We assume that a recovered node can not be reinfected again. Thus a node that is isolated on any day $t$ has no impact on the network from then onwards. Such nodes can be regarded as "removed". Therefore, it is removed from the contact network for all subsequent times $t, t+1, \ldots$. Fig. 1 (right) depicts a contact network at a given time $t$. We assume that a public health authority knows the entire contact network and decides who to test based on this information. This assumption has been made in several other works in this genre eg in [38].

Denote the set of neighbors of node $i$, in day $t$, by $\partial_i(t)$. The state of each node at time $t+1$ depends on the state of its neighbors $\partial_i(t)$, as well as its own state in day $t$, as given by the following conditional probability:

$$\Pr\left(\sigma_i(t+1)|\{\sigma_j(t)\}_{j\in\partial_i^+(t)}\right)$$

where $\partial_i^+(t) = \partial_i(t) \cup \{i\}$.

Node $i$ is tested positive on day $t$ if it is in the infectious state $(I)$[1]. Let $Y_i(t)$ denote the test result:

$$Y_i(t) = \left\{ \begin{array}{cc} 1 & \sigma_i(t) = I \\ 0 & \sigma_i(t) \in \{S, L, R\}. \end{array} \right. \tag{1}$$

We do not assume any type of error in testing and $Y_i(t)$ is hence a deterministic function of $\sigma_i(t)$. Let $\mathcal{O}(t)$ be the set of nodes that have been tested (observed) in day $t$ and denote the *network observations* at time $t$ by $\underline{Y}(t) = \{Y_i(t)\}_{i\in\mathcal{O}(t)}$.

Our goal in this paper is to design testing and isolation strategies in order to contain the spread and minimize the cumulative infections. Naturally, testing resources (and hence observations) are often limited and such constraints make decision making challenging. Let $B(t)$ be the maximum number of tests that could be performed on day $t$, called the *testing budget*. $B(t)$ can evolve based on the system necessities, e.g., in contact tracing that is widely deployed for COVID-19, the number of tests is chosen based on the history of observations[2]. Also, governments often upgrade testing infrastructure as the number of cases increase. Our framework captures both fixed and time-dependent budget $B(t)$, but we focus on time-dependent $B(t)$ for simulations.

Define the cumulative infections on day $t$, denoted by $C^\pi(t)$, as the number of nodes who have been infected before and including day $t$, where $\pi$ is the testing and isolation policy. Let $\mathcal{K}^\pi(t)$ denote the set of tests $\pi$ performs on day $t$. Given a large time horizon $T$, our objective is:

$$\begin{aligned} \min_{\pi} \quad & \mathbb{E}[C^\pi(T)] \\ s.t. \quad & |\mathcal{K}^\pi(t)| \leq B(t),\, 0 \leq t \leq T-1. \end{aligned} \tag{2}$$

Recall that $\sigma_i(t)$, the state of node $i$ on day $t$, is a random variable and unknown. For each node $i$, define a probability vector $\underline{v}_i(t)$ of size $|\mathcal{X}|$, where each coordinate is the probability of the node being in a particular state at the end of time $t$. The coordinates of $\underline{v}_i(t)$ follow the order $(I, L, R, S)$ and we have

$$\underline{v}_i(t) = \left[v_x^{(i)}(t)\right]_{x\in\mathcal{X}}, \qquad v_x^{(i)}(t) = \Pr\left(\sigma_i(t) = x\right). \tag{3}$$

For example, $v_S^{(i)}(t)$ represents the probability of node $i$ being in state $S$ in time $t$. We now define $F_i(\mathcal{D}; t)$ to be the *conditional* probability of node $i$ being infected by nodes in $\mathcal{D}$ (for the first time) at day $t$, as a function of the nodes' states $\{\sigma_i(t)\}_{i\in\mathcal{V}(t)}$. We have

$$F_i(\mathcal{D}; t) = 1_{\{\sigma_i(t)=S\}} \cdot \prod_{j\in\partial_i(t)\backslash\mathcal{D}} \left(1 - \beta 1_{\{\sigma_j(t)=I\}}\right) \cdot \left(1 - \prod_{j\in\mathcal{D}\cap\partial_i(t)} \left(1 - \beta 1_{\{\sigma_j(t)=I\}}\right)\right). \tag{4}$$

---

[1]We assume that a node in the latent state $L$ is infected, but not infectious. We further assume that latent nodes test negative.

[2]In practical implementations, scheduling constraints do play a role but we disregard that in this work.

Equation (4), captures the impact that the nodes in $\mathcal{D}$ have on infecting node $i$ at day $t$. In this equation we assume that the infections from different nodes are independent. The same assumption has also been made in several other papers in this genre, eg in [27, 28, 29]. Then, we find the expectation (with respect to $\{\sigma_i(t)\}_{i \in \mathcal{V}(t)}$) of (4) as follows:

$$\mathbb{E}_{\{\sigma_i(t)\}_{i \in \mathcal{V}(t)}} [F_i(\mathcal{D}; t)] = v_S^{(i)}(t) \cdot \Big\{ \prod_{j \in \partial_i(t) \backslash \mathcal{D}} (1 - \beta v_I^{(j)}(t)) \Big\} \cdot \Big\{ 1 - \prod_{j \in \mathcal{D} \cap \partial_i(t)} (1 - \beta v_I^{(j)}(t)) \Big\}. \tag{5}$$

It is worth noting that (4) is a probability conditioned on $\{\sigma_i(t)\}_{i \in \mathcal{V}(t)}$, while (5) is an unconditional probability. To obtain (5), we have indeed assumed that the states of the nodes are independent. This assumption does not hold in general and we only utilize it here to obtain a simple expression in (5) in terms of the infection probabilities. We do not use this independence assumption in the rest of the paper. Define

$$S(\mathcal{D}; t) = \sum_{i \in \mathcal{V}(t)} \mathbb{E}[F_i(\mathcal{D}; t)]. \tag{6}$$

Here, $S(\mathcal{D}; t)$ represents the (expected) number of newly infectious nodes incurred by nodes in $\mathcal{D}$ at day $t$. Recall that $\mathcal{K}^\pi(t)$ be the set of nodes that are tested at time $t$. We show the following result in Appendix A.

**Lemma 1.** $\mathbb{E}[C^\pi(t+1) - C^\pi(t)] = S(\mathcal{V}(t) \backslash \mathcal{K}^\pi(t); t)$.

## 2.1 Supermodularity

It is complex to solve (2) globally, especially if one seeks to find solutions that are optimal looking into the future. We thus simplify the optimization (2) for policies that are myopic in time as follows. First, note that $C^\pi(T)$ can be re-written as follows through a telescopic sum:

$$C^\pi(T) = \sum_{t=0}^{T-1} C^\pi(t+1) - C^\pi(t). \tag{7}$$

Then, we restrict attention to myopic policies that at each time minimize $\mathbb{E}[C^\pi(t+1) - C^\pi(t)]$. We then show how $\mathbb{E}[C^\pi(t+1) - C^\pi(t)]$ can be expressed in terms of a supermodular function.

Using (7) along with Lemma 1, we seek to solve the following optimization sequentially in time for $0 \leq t \leq T - 1$:

$$\min_{|\mathcal{K}^\pi(t)| \leq B(t)} S(\mathcal{V}(t) \backslash \mathcal{K}^\pi(t); t). \tag{8}$$

We now prove some desired properties for the set function $S(\mathcal{K}^\pi(t); t)$ (see Appendix B).

**Theorem 1.** $S(\mathcal{K}^\pi(t); t)$ defined in (6) is a supermodular[3] and increasing monotone function on $\mathcal{K}^\pi(t)$.

On day $t$, and given the network, the probability vectors of all nodes, and $\mathcal{K}_1^\pi(t) \subset \mathcal{K}_2^\pi(t)$, for any node $i \notin \mathcal{K}_2^\pi(t)$, node $i$ will incur larger increment of newly infectious nodes under $\mathcal{K}_2^\pi(t)$ than that under $\mathcal{K}_1^\pi(t)$. This is because node $i$ may have common neighbors with nodes in $\mathcal{K}_2^\pi(t)$. So, supermodularity holds in Theorem 1.

The optimization (8) is NP-hard [59]. However, using the supermodularity of $S(\mathcal{V}(t) \backslash \mathcal{K}^\pi(t); t)$, we propose Algorithm 1 based on [60, Algorithm A] to greedily optimize (8) in every day $t$. Denote the optimum solution of (8) as OPT. As proved in [60], on every day $t$, Algorithm 1 attains a solution, denoted by $\tilde{\mathcal{K}}^\pi(t)$, such that $(\mathcal{V}(t) \backslash \tilde{\mathcal{K}}^\pi(t); t) \leq (1 + \epsilon(t)) \cdot$ OPT, i.e., the solution $\tilde{\mathcal{K}}^\pi(t)$ is an $\epsilon(t)$-approximation of the optimum solution. Here, on day $t$, the constant $\epsilon(t)$, which is the steepness of the set function $S(\cdot; t)$ as described in [60], can be calculated as follows, $\epsilon(t) = \frac{\epsilon'}{4(1-\epsilon')}$ and $\epsilon' = \max_{a \in \mathcal{V}(t)} \frac{S(\mathcal{V}(t); t) - S(\mathcal{V}(t) \backslash \{a\}; t) - S(\{a\}; t)}{S(\mathcal{V}(t); t) - S(\mathcal{V}(t) \backslash \{a\}; t)}$.

In Algorithm 1, on every day $t$, in every step, we choose the node who provides the minimum increment on $S(\cdot; t)$ based on the results in the previous step, and then remove the node from the current node set. Algorithm 1 is stopped when $K^\pi(t)$ nodes have been chosen. The complexity of this algorithm is discussed in Appendix C.

---

[3]Let $\mathcal{X}$ be a finite set. A function $f : 2^{\mathcal{X}} \to \mathbb{R}$ is supermodular if for any $\mathcal{A} \subset \mathcal{B} \subset \mathcal{X}$, and $x \in \mathcal{X} \backslash \mathcal{B}$, $f(\mathcal{A} \cup \{x\}) - f(\mathcal{A}) \leq f(\mathcal{B} \cup \{x\}) - f(\mathcal{B})$.

---

**Algorithm 1** Greedy Algorithm

---

    **Step 0**: On day $t$, input $\{\underline{v}_i(t)\}_{i \in \mathcal{V}(t)}$, set $\mathcal{A}_0 = \mathcal{V}(t)$.
   **repeat**
      **Step i**: Let $\mathcal{A}_i = \mathcal{A}_{i-1} \backslash \{a_i\}$, where

$$a_i = \arg \min_{a \in \mathcal{A}_{i-1}} S\big(\{a\} \cup \{a_1, \cdots, a_{i-1}\}; t\big).$$

   **until** $i = N(t) - |\mathcal{K}^\pi(t)|$, and return $\mathcal{K}^\pi(t) = \mathcal{A}_i$.

---

## 3 Exploitation and Exploration

In Section 2.1, we proposed a near-optimal greedy algorithm to sequentially (in time) select the nodes to test. However, Algorithm 1 has two shortcomings. (i) The computation is costly when $N$ and/or $T$ are large (see Appendix C). (ii) The objective function $S\big(\mathcal{V}(t) \backslash \mathcal{K}^\pi(t)\big)$ is dependent on $\{\underline{v}_i(t)\}_{i \in \mathcal{V}(t)}$ which is unknown, even though the network and the process are stochastically fully given (see Section 2). This is because the set of infected nodes are unknown and time-evolving.

To overcome the first shortcoming, we propose a simpler reward maximization policy by minimizing an upper bound on the objective function in (8). To overcome the second shortcoming, we estimate $\{\underline{v}_i(t)\}_{i \in \mathcal{V}(t)}$ using the history of test observations $\{\underline{Y}(\tau)\}_{\tau=0}^{t}$ (as presented in Section 4). we refer to the estimates as $\{\underline{u}_i(t)\}_{i \in \mathcal{V}(t)}$. Both the greedy policy and its reward-based variant that we will propose in this section thus need to perform decision making based on the estimates $\{\underline{u}_i(t)\}_{i \in \mathcal{V}(t)}$ and we refer to them as "exploitation" policies.

It now becomes clear that testing has two roles: to find the infected in order to isolate them and contain the spread, and to provide better estimates of $\{\underline{v}_i(t)\}_{i \in \mathcal{V}(t)}$. This leads to interesting tradeoffs between exploitation and exploration as we will discuss next. Under exploitation policies, we test nodes deterministically based on a function of $\{\underline{v}_i(t)\}_{i \in \mathcal{V}(t)}$, (which is called "reward", and will be defined later); while under exploration policies, nodes are tested according to a probabilistic framework (based on rewards of all nodes).

To simplify the decision making into reward maximization, we first derive an upper bound on $S\big(\mathcal{V}(t) \backslash \mathcal{K}^\pi(t); t\big)$. Define

$$r_i(t) = S\big(\{i\}; t\big). \tag{9}$$

Using the supermodularity of the function $S(\cdot)$, we prove the following lemma in Appendix D.

**Lemma 2.** $S\big(\mathcal{V}(t) \backslash \mathcal{K}^\pi(t); t\big) \leq S\big(\mathcal{V}(t); t\big) - \sum_{i \in \mathcal{K}^\pi(t)} r_i(t).$

**Remark 1.** *Recall that $S(\cdot; t)$ is a supermodular function, then the amount of newly infectious nodes incurred by the set $\mathcal{K}^\pi(t)$, $S(\mathcal{K}^\pi(t); t)$, is larger than the sum of the amount of newly infectious nodes by every individual node in $\mathcal{K}^\pi(t)$, i.e., $\sum_{i \in \mathcal{K}^\pi(t)} r_i(t)$. Thus, $S\big(\mathcal{V}(t) \backslash \mathcal{K}^\pi(t); t\big)$ is upper bounded by $S\big(\mathcal{V}(t); t\big) - \sum_{i \in \mathcal{K}^\pi(t)} r_i(t)$.*

We propose to minimize the upper bound in Lemma 2 instead of $S\big(\mathcal{V}(t) \backslash \mathcal{K}^\pi(t); t\big)$. Since $\mathcal{V}(t)$ is known and $S\big(\mathcal{V}(t); t\big)$ is hence a constant, the problem reduces to solving:

$$\max_{|\mathcal{K}^\pi(t)| \leq B(t)} \sum_{i \in \mathcal{K}^\pi(t)} r_i(t). \tag{10}$$

Given probabilities $\{\underline{v}_i(t)\}_{i \in \mathcal{V}(t)}$, the solution to (10) is to pick the nodes associated with the $B(t)$ largest values $r_i(t)$. We thus refer to $r_i(t)$ as the *reward* of selecting node $i$.

Let $\{\underline{u}_i(t)\}_{i \in \mathcal{V}(t)}$ be an estimate for $\{\underline{v}_i(t)\}_{i \in \mathcal{V}(t)}$ found by estimating the conditional probability of the state of node $i$ given the history of observations $\{\underline{Y}(\tau)\}_{\tau=0}^{t-1}$. Our proposed reward-based Exploitation (RbEx) policy follows the same idea of selecting the nodes with the highest rewards. Note that $\{\underline{v}_i(t)\}_{i \in \mathcal{V}(t)}$ is unknown to all nodes. Instead of using the true probabilities $\{\underline{v}_i(t)\}_{i \in \mathcal{V}(t)}$, we consider the estimates of it which

we sequentially update by computing the prior probabilities $\{\underline{u}_i(t)\}_{i \in \mathcal{V}(t)}$ and the posterior probabilities $\{\underline{w}_i(t)\}_{i \in \mathcal{V}(t)}$. In particular, $\{\underline{u}_i(0)\}_{i \in \mathcal{V}(0)}$ and $\{\underline{w}_i(0)\}_{i \in \mathcal{V}(0)}$ are the prior probabilities and the posterior probabilities on the initial day, respectively. Hence, we calculate the estimate of rewards, denoted by $\hat{r}_i(t)$, by replacing $\{\underline{v}_i(t)\}_{i \in \mathcal{V}(t)}$ with $\{\underline{u}_i(t)\}_{i \in \mathcal{V}(t)}$ in (6) and (9).

---

**Algorithm 2** Reward-based Exploitation (RbEx) Policy

---

Input $\{\underline{w}_i(0)\}_{i \in \mathcal{V}(t)}$, $\{\underline{u}_i(0)\}_{i \in \mathcal{V}(0)}$, $\underline{Y}(0)$, and $t = 0$.
**Repeat** for $t = 1, 2, \cdots, T-1$.
**Step 1**: Calculate $\{\hat{r}_i(t)\}_{i \in \mathcal{V}(t)}$ based on $\{\underline{u}_i(t)\}_{i \in \mathcal{V}(t)}$ and (9).
**Step 2**: Re-arrange the sequence $\{\hat{r}_i(t)\}_{i \in \mathcal{V}(t)}$ in descending order, and test the first $B(t)$ nodes. Get the new observations $\underline{Y}(t)$.
**Step 3**: Based on $\underline{Y}(t)$, update $\{\underline{u}_i(t+1)\}_{i \in \mathcal{V}(t+1)}$ by Algorithm 4 (Step 0 $\sim$ Step 2) in Section 4.

---

The shortcoming of Algorithm 2 is that it targets maximizing the estimated sum rewards, even though the estimates may be inaccurate. In this case, testing is heavily biased towards the history of testing and it does not provide opportunities for getting better estimates of the rewards. For example, consider a network with several clusters. If one positive node is known by Algorithm 2, then it may get stuck in that cluster and fail to locate more positives in other clusters.

In Section 4.2, we will prove, in a line network, that the exploitation policy described in Algorithm 2 can be improved by a constant factor (in terms of the resulting cumulative infections) if a simple form of exploration is incorporated.

We next propose an exploration policy. Our proposed policy is probabilistic in the sense that the nodes are randomly tested with probabilities that are proportional to their corresponding estimated rewards. This approach has similarities and differences to Thompson sampling and more generally posterior sampling. The similarity lies in the probabilistic nature of testing using posterior probabilities. The difference is that in our setting decision making depends on the distributions of decision variables, but not samples of the decision variables.

More specifically, at time $t$, node $i$ is tested with probability $\min\{1, \frac{B(t)\hat{r}_i(t)}{\sum_{j \in \mathcal{V}(t)} \hat{r}_j(t)}\}$, which depends on the budget $B(t)$. Note that each node is tested with probability at most 1; so if $\frac{B(t)\hat{r}_i(t)}{\sum_{j \in \mathcal{V}(t)} \hat{r}_j(t)} > 1$ for some node $i$, then we would not fully utilize the budget. The unused budget is thus

$$c(t) = \sum_{i \in \mathcal{V}(t)} \left(\frac{B(t)\hat{r}_i(t)}{\sum_{j \in \mathcal{V}(t)} \hat{r}_j(t)} - 1\right)^+ \tag{11}$$

and can be used for further testing[4]. Algorithm 3 outlines our proposed Reward-based Exploitation-Exploration (REEr) policy.

---

**Algorithm 3** Reward-based Exploitation-Exploration (REEr) Policy

---

Input $\{\underline{w}_i(0)\}_{i \in \mathcal{V}(t)}$, $\{\underline{u}_i(0)\}_{i \in \mathcal{V}(0)}$, $\underline{Y}(0)$, and $t = 0$.
**Repeat** for $t = 1, 2, \cdots, T-1$
**Step 1**: Calculate $\{\hat{r}_i(t)\}_{i \in \mathcal{V}(t)}$ based on $\{\underline{u}_i(t)\}_{i \in \mathcal{V}(t)}$ and (9).
**Step 2**: Test node $i$ with probability $\min\{1, \frac{B(t)\hat{r}_i(t)}{\sum_{j \in \mathcal{V}(t)} \hat{r}_j(t)}\}$. After that, randomly select $c(t)$ (defined in (11)) further nodes to test (see Footnote 4). Get the new observations $\underline{Y}(t)$.
**Step 3**: Based on $\underline{Y}(t)$, update $\{\underline{u}_i(t+1)\}_{i \in \mathcal{V}(t+1)}$ by Algorithm 4 (Step 0 $\sim$ Step 2) in Section 4.

---

[4]Note that $c(t)$ is not always an integer. Instead of $c(t)$, we use $\texttt{Int}\big(c(t)\big)$ with probability $|\texttt{Int}\big(c(t)\big) - c(t)|$ where $\texttt{Int}(\cdot) \in \{\lfloor\cdot\rfloor, \lceil\cdot\rceil\}$.

## 4 Message-Passing Framework

As discussed in Section 3, the probabilities $\{\underline{v}_i(t)\}_i$ are unknown. In this section, we develop a message passing framework to sequentially estimate $\{\underline{v}_i(t)\}_i$ based on the network observations and the dynamics of the spread process. We refer to these estimates as $\{\underline{u}_i(t)\}_i$.

When node $i$ is tested on day $t$, an observation $Y_i(t)$ is provided about its state. Knowing the state of node $i$ provides two types of information: (i) it provides information about the state of the neighboring nodes in future time slots $t+1, t+2, \ldots$ (because of the evolution of the spread in time and on the network), and (ii) it also provides information about the past of the spread, meaning that we can infer about the state of the (unobserved) nodes at previous time slots. For example, if node $i$ is tested positive in time $t$, we would know that (i) its neighbors are more likely to be infected in time $t+1$ and (ii) some of its neighbors must have been infected in a previous time for node $i$ to be infected now. This forms the basis for our backward-forward message passing framework.

Given the spread model of Section 2, we first describe the forward propagation of belief. Suppose that at time $t$, the probability vector $\underline{v}_i(t)$ is given for all $i$. The probability vector $\underline{v}_i(t+1)$ can be computed as follows (see Appendix E):

$$\underline{v}_i(t+1) = \underline{v}_i(t) \times \mathtt{P}_i\big(\{\underline{v}_j(t)\}_{j \in \partial_i^+(t)}\big) \tag{12}$$

where $\mathtt{P}_i\big(\{\underline{v}_j(t)\}_{j \in \partial_i^+(t)}\big)$ is a local transition probability matrix given in Appendix E.

Recall that $\underline{Y}(t)$ denotes the collection of network observations on day $t$. The history of observations is then denoted by $\{\underline{Y}(\tau)\}_{\tau=1}^{t-1}$. Based on these observations, we wish to find an estimate of the probability vector $\underline{v}_i(t)$ for each $i \in \mathcal{V}(t)$. We denote this estimate by $\underline{u}_i(t) = (u_x^{(i)}(t), x \in \mathcal{X})$ and refer to it, in this section, as the *prior probability* of node $i$ in time $t$. We further define the *posterior probability* $\underline{w}_i(t) = (w_x^{(i)}(t), x \in \mathcal{X})$ of node $i$ in time $t$ (after obtaining new observations $\underline{Y}(t)$). In particular,

$$u_x^{(i)}(t) = \Pr\big(\sigma_i(t) = x | \{\underline{Y}(\tau)\}_{\tau=1}^{t-1}\big)$$
$$w_x^{(i)}(t) = \Pr\big(\sigma_i(t) = x | \{\underline{Y}(\tau)\}_{\tau=1}^{t}\big).$$

Here, the prior probability is defined *at the beginning of* every day, and the posterior probability is defined *at the end of* every day. Conditioning all probabilities in (12) on $\{\underline{Y}(\tau)\}_{\tau=1}^{t}$, we obtain the following *forward-update rule* (see Appendix F)

$$\underline{u}_i(t+1) = \underline{w}_i(t) \times \mathtt{P}_i\big(\{\underline{w}_j(t)\}_{j \in \partial_i^+(t)}\big). \tag{13}$$

**Remark 2.** *Following (13), we need to utilize the observations $\underline{Y}(t)$ and the underlying dependency among nodes' states to update the posterior probabilities $\{\underline{w}_i(t)\}_i$, and consequently update $\{u_i(t+1)\}_i$ based on the forward-update rule (13). This is however non-trivial. A Naive approach would be to locally incorporate node $i$'s observation $Y_i(t)$ into $w_i(t)$ and obtain $\underline{u}_i(t+1)$ using (13). This approach, however, does not fully exploit the observations and it disregards the dependency among nodes' states, as caused by the nature of the spread (An example is provided in Appendix H).*

**Backward Propagation of Belief**   To capture the dependency of nodes' states and thus best utilize the observations, we proceed as follows. First, denote

$$\underline{e}_i(t-1) = (e_x^{(i)}(t-1), x \in \mathcal{X})$$
$$e_x^{(i)}(t-1) = \Pr\big(\sigma_i(t-1) = x | \{\underline{Y}(\tau)\}_{\tau=1}^{t}\big).$$

Vector $\underline{e}_i(t-1)$ is the posterior probability of node $i$ at time $t-1$, after obtaining the history of observations up to and including time $t$. By computing $\underline{e}_i(t-1)$, we are effectively correcting our belief on the state of the nodes in the previous time slot by *inference* based on the observations acquired at time $t$. This constitutes the *backward step* of our framework and we will expand on it shortly. The backward step can be repeated to correct our belief also in times $t-2$, $t-3$, etc. For clarity of presentation and tractability of our analysis

and experiments, we truncate the backward step at time $t - 1$ and present assumptions under which this truncation is theoretically justifiable. Considering larger truncation windows is straightforward but out of the scope of this paper.

Once our belief about nodes' states is updated in prior time slots (e.g., $\underline{e}_i(t-1)$ is obtained), it is propagated forward in time for *prediction* and to provide a more accurate estimate of the nodes' posterior and prior probabilities. More specifically, consider (12) written for time $t$ (rather than $t + 1$) and condition all probabilities on $\{\underline{Y}(\tau)\}_{\tau=1}^t$. We obtain the following update rule (see Appendix F):

$$\underline{w}_i(t) = \underline{e}_i(t-1) \times \tilde{\mathrm{P}}_i\big(\{\underline{e}_j(t-1)\}_{j \in \partial_i^+(t-1)}\big) \tag{14}$$

where $\tilde{\mathrm{P}}_i\big(\{\underline{e}_j(t-1)\}_{j \in \partial_i^+(t-1)}\big)$ is given in Appendix F. *Note that the local transition matrix in (14) is not the same as (13). This is because "future" observations were available in* $\tilde{\mathrm{P}}_i\big(\{\underline{e}_j(t-1)\}_{j \in \partial_i^+(t-1)}\big)$. The probability vectors $\{\underline{e}_i(t-1)\}_i$ provide better estimates for $\{\underline{w}_i(t)\}_i$ through (14) and the prior probabilities $\{\underline{u}_i(t+1)\}_i$ are then computed using (13) to be used for decision making in time $t + 1$. The block diagram in Fig. 3 depicts the high-level idea of our framework. It is worth noting that $\mathrm{P}_i\big(\{\underline{w}_j(t)\}_{j \in \partial_i^+(t)}\big)$ in (13) and $\tilde{\mathrm{P}}_i\big(\{\underline{e}_j(t-1)\}_{j \in \partial_i^+(t-1)}\big)$ in (14) both depend on the observations, $\{\underline{Y}(\tau)\}_{\tau=1}^t$.

We next discuss how $\underline{e}_j(t-1)$ can be computed, starting with some notations. Denote by

$$\zeta_i(t) = \sigma_i(t)|_{\{\underline{Y}(\tau)\}_{\tau=1}^t}, \quad \theta_j(t) = \sigma_i(t)|_{\{\underline{Y}(\tau)\}_{\tau=1}^{t-1}}, \tag{15}$$

the state of the nodes in the posterior probability spaces conditioned on the observations $\{\underline{Y}(\tau)\}_{\tau=1}^t$ and $\{\underline{Y}(\tau)\}_{\tau=1}^{t-1}$, respectively. We further define $\Psi_i(t)$ to be the set of those neighbors of node $i$ at time $t - 1$, including node $i$, who are observed/tested at time $t$. This set consists of all nodes whose posterior probabilities will be updated at time $t - 1$ (given a new observation $Y_i(t)$). The set of all neighbors (except node $i$) of the nodes in $\Psi_i(t)$ then defines $\Phi_i(t)$. The set $\Phi_i(t)$ consists of all nodes whose posterior probabilities at time $t$ is updated by the observation $Y_i(t)$. More precisely, we have

$$\Psi_i(t) = \mathcal{O}(t) \cap \partial_i^+(t-1),$$
$$\Phi_i(t) = \{j | j \in \partial_k^+(t-1), k \in \Psi_i(t)\} \backslash \{i\},$$
$$\Theta_i(t) = \{j | j \in \partial_k^+(t-1), k \in \mathcal{O}(t)\} \backslash \{i\}$$

where $\mathcal{O}(t)$ is the set of observed nodes at time $t$ (see Figure 2). In Appendix G, we show

$$e_x^{(i)}(t-1) = \frac{\Pr\big(\underline{Y}(t)|\zeta_i(t-1) = x\big) \ w_x^{(i)}(t-1)}{\Pr\big(\underline{Y}(t)\big)}. \tag{16}$$

It suffices to find $\Pr\big(\underline{Y}(t)|\zeta_i(t-1) = x\big)$. The denominator $\Pr\big(\underline{Y}(t)\big)$ is then found by normalization of the enumerator in (16). Let $\{x_j\}_{j \in \mathcal{O}(t)}$ be a realization of $\{\theta_j(t)\}_{j \in \mathcal{O}(t)}$ and $\{y_l\}_{l \in \Theta_i(t)}$ be a realization of $\{\zeta_l(t-1)\}_{l \in \Theta_i(t)}$. We prove the following in Appendix G under a simplifying truncation assumption (see Assumption 1 in Appendix G) where the backward step is truncated in time $t - 1$:

$$\Pr\big(\underline{Y}(t)|\zeta_i(t-1) = x\big) = \Pr\big(\{Y_j(t)\}_{j \in \Psi_i(t)}|\zeta_i(t-1) = x\big)$$
$$= \sum_{\{x_j\}_{j \in \Psi_i(t)}} \prod_{j \in \Psi_i(t)} \Pr\big(Y_j(t)|\theta_j(t)\big)$$
$$\times \sum_{\{y_l\}_{l \in \Phi_i(t)}} \prod_{j \in \Psi_i(t)} \Pr\big(x_j|\{y_l\}_{l \in \partial_j^+(t-1) \backslash \{i\}}, x\big) \times \prod_{l \in \{\Phi_i(t)\}} w_{y_l}^{(i)}(t-1). \tag{17}$$

We finally present our *Backward-Forward Algorithm* to sequentially compute estimates $\{\underline{u}_i(t)\}_i$ in Algorithm 4. The process of Algorithm 4 is given in Fig 3, and we also give a simple example to show the process of Algorithm 4 in Appendix H.

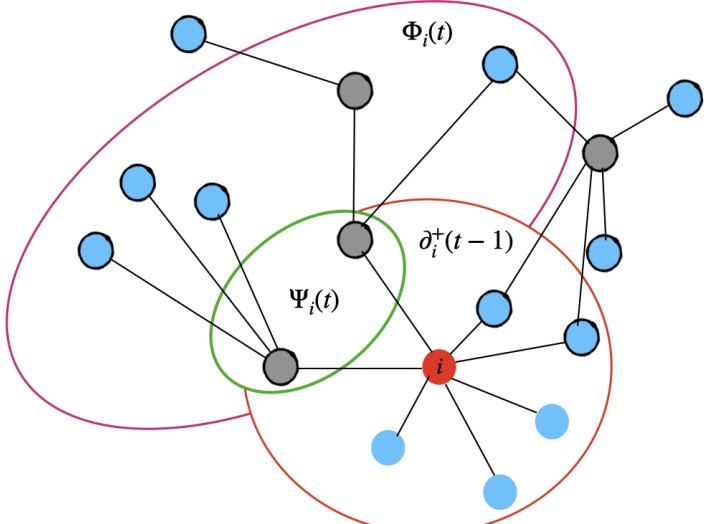

Figure 2: An example of $\Psi_i(t)$, $\Phi_i(t)$ and $\Theta_i(t)$. Node $i$ is marked in red, and its neighborhood $\partial_i^+(t-1)$ is shown by the red contour. Suppose that the gray nodes are tested on day $t-1$, then $\Psi_i(t)$ is the set of nodes within the green contour, and $\Phi_i(t)$ consists of the nodes in the purple contour. Finally, nodes in $\Theta_i(t)$ are marked with bold black border

.

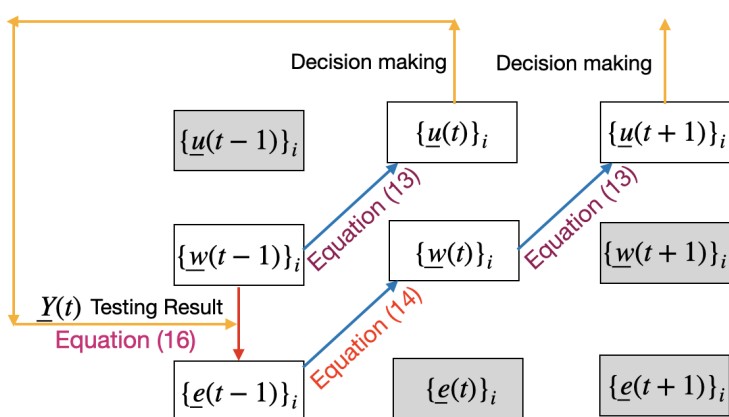

Figure 3: The process of Algorithm 4. One example of a complete process is given in unshaded blocks. Recall that $\underline{u}_i(\tau)$, $\underline{w}_i(\tau)$, and $\underline{e}_i(\tau)$, where $\tau \in \{t-1, t, t+1\}$, are the prior probabilities, the posterior probabilities, and the updated posterior probabilities, respectively.

## 4.1 Necessity of Backward Updating

Now we provide an example which illustrates the necessity of backward updating.

---

**Algorithm 4** Backward-Forward Algorithm

---

Input $\underline{Y}(0)$, $\{\underline{e}_i(0)\}_{i \in \mathcal{V}(0)}$, $\{\underline{w}_i(0)\}_{i \in \mathcal{V}(0)}$, $\{\underline{u}_i(0)\}_{i \in \mathcal{V}(0)}$.
**Repeat** for $t = 1, 2, \cdots, T-1$
**Step 0**: Based on $\underline{Y}(t)$, get $\mathcal{V}(t)$ from $\mathcal{V}(t-1)$.
**Step 1**: Backward step. Update $\underline{e}_i(t-1)$ by (16), (17), and then compute $\underline{w}_i(t)$ by (14).
**Step 2**: Forward step. Compute $\underline{u}_i(t+1)$ by (13).

---

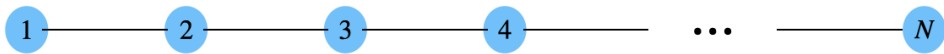

Figure 4: The line network in Example 1.

**Example 1.** *Consider a line network with the node set $\mathcal{V} = \{1, 2, \ldots, N\}$ and the edge set $\mathcal{E} = \{(i, i+1), 1 \leq i \leq N-1\}$ (see Figure 4). On the initial day, we assume that each node is infected independently with probability $1/N$. Let $\beta = 1$, $\lambda = 0$, $\gamma = 0^5$, and $B(t) = 1$. We further assume that there is no isolation when a positive node is tested.*

Based on Example 1, we show that the naive approach of Remark 2 (i.e., forward-only updating) will cause the estimated probabilities to never converge to the true probabilities of infection. Nonetheless, if we use the Backward-Forward Algorithm 4, the estimated probabilities converge to the true probabilities after a certain number of steps. Formally, we prove the following result in Appendix J.

**Theorem 2.** *For any testing policy that sequentially computes $\{\underline{u}_i(t)\}_i$ based on (13) (see Remark 2), with probability (approximately) $\frac{1}{e}$, we have $\sum_{i=1}^{N} ||\underline{v}_i(t) - \underline{u}_i(t)|| \overset{t \to \infty}{\to} \Theta(N)$, for large $N$ [6]. On the other hand, there exists a testing policy that sequentially updates $\{\underline{u}_i(t)\}_i$ based on Algorithm 4 and attains $\sum_{i=1}^{N} ||\underline{v}_i(t) - \underline{u}_i(t)|| = 0$, $t \geq 2N$.*

*Roadmap of proof: Consider a simple case where every node is susceptible. Since each node is infected with probability $1/N$, then the case occurs with probability $\simeq 1/e$.*

*Under the case above, consider any testing policy based on the algorithm in Remark 2. If a node is tested on day $t$, then the policy "clears" the tested node. Since the updating rule of the algorithm can not go back to the information on day $t-1$, then it can not "clear" any neighbors of the tested node and its probability of infection updates to a non-zero value in the next day. Furthermore, we show that almost all nodes have an significantly large probability of infection when time horizon is sufficiently large, hence $\sum_{i=1}^{N} ||\underline{v}_i(t) - \underline{u}_i(t)|| \overset{t \to \infty}{\to} \Theta(N)$.*

*On the other hand, we can propose a specific testing policy. Note that there is no infection, if Algorithm 4 is used to update probabilities, then it can reveal the states of all nodes under the specific testing policy after at most $2N$ days. So we have $\sum_{i=1}^{N} ||\underline{v}_i(t) - \underline{u}_i(t)|| = 0$, $t \geq 2N$.*

In Theorem 2, we illustrate the necessity of backward updating when testing is limited. In essence, we want to "clear" the graph and confirm that there are no infections. If the number of tests is limited, we have mathematically shown that no algorithm can correctly estimate the nodes' infection probabilities if it does not use the backward (inference) step. On the contrary, there is an algorithm that uses the backward step along with the forward step and the estimates that it provides for the nodes' infection probabilities converge to the true probabilities of the nodes after some finite steps. Even though the considered graph is simple but the phenomena it captures is general.

As discussed in Theorem 2, the backward updating is necessary. However, bacward updating can be computationally expensive in large dense graphs. To trade off the impact of backward updating and the reduction of computation complexity, we propose an $\alpha$-linking backward updating algorithm in Appendix K, where Algorithm 4 is applied on a random subgraph with fewer edges.

## 4.2 Necessity of Exploration

Note that in reality we have no information for $\{\underline{v}_i(t)\}_i$, and only have the estimates $\{\underline{u}_i(t)\}_i$. One may wonder if exploitation based on wrong initial estimated probability vectors, i.e., $\{\underline{u}_i(0)\}_i$, misleads decision making by providing poorer and poorer estimates of the probabilities of infection. If so, exploration may be necessary.

---

[5]Here, $\lambda = 0$ implies there is no latent state, and $\gamma = 0$ implies that nodes never recover.

[6]Theorem 2 holds for all kinds of noem due to the equivalence of norms. In addition, the convergence is topological convergence.

**Example 2.** *Consider $\mathcal{V} = \{1, 2, \ldots, N\}$ and edges $\mathcal{E} = \{(i, i+1), 1 \leq i \leq N-1\}$ (see Figure 4). Let $N \gg 10$, $\beta = 1$, $\lambda = 0$, $\gamma = 0$, and $B(t) = 10$. Suppose that on the initial day, node 1 is infected and all other nodes are susceptible. Consider a wrong initial estimate: $w_I^{(i)}(0) = u_I^{(i)}(0) = 0$ if $i \leq \frac{9N}{10}$, and $w_I^{(i)}(0) = u_I^{(i)}(0) = \frac{10\epsilon}{N}$ otherwise, where $\epsilon > 0$. With this initial belief, we have $\sum_{i=1}^{N} ||\underline{w}_i(0) - \underline{v}_i(0)|| = O(1 + \epsilon)$.*

Different from Example 1, here we consider the isolation of nodes that are tested positive. In Example 2, suppose that a specific exploration policy is applied: 1 (out of 10) tests is done randomly, and the other 9 tests are done following exploitation. Now, in Appendix L, we show that under the RbEx policy, the cumulative infection is at least $aN$ for a constant $a$, while under the exploration policy defined above, the cumulative infection is at most $bN$ with very high probability, and the ratio $a/b$ can be any constant for a large enough $N$. More formally, we have the following theorem. Let $p_0$ be a large probability and consider a large time horizon $T$. Denote the cumulative infections under the RbEx policy by $C^{RbEx}(T)$ and under the specific exploration policy defined above by $C^{exp}(T)$. We prove the necessity of exploration in the following Theorem.

**Theorem 3.** *With probability $p_0 \geq \frac{99}{100}$, $\frac{C^{RbEx}(T)}{C^{exp}(T)} \geq c(N, p_0)$, where $c(N, p_0)$ is a constant only depending on $N$ and $p_0$.*

*Roadmap of proof: Under the RbEx policy, we test nodes based on their predicted probabilities. Since the nodes that are located towards the end of the line (right side in Fig. 4) have non-zero probabilities, they are tested first while the disease spreads on the other end of the network (left side in Fig. 4). Mathematically, suppose that for the first time, an infectious node is tested at day $t = aN$, then there are at least $\min\{aN, N\}$ infectious nodes before the spread can be contained.*

*Under the specific exploration policy described above, consider the event that, for the first time, an infectious node is explored on day $t = b'N$ ($b' < a$). We argue that with probability $p_0$, the exploration policy catches at least two new infections at each step after $t = b'N$. After $2t$, the algorithm catches all the infections, and we have at most $2b'N$ infections. Let $b = 2b'$. This is an improvement by a factor of at least $\frac{a}{b}$ in comparison to the RbEx strategy. Factor $\frac{a}{b}$ depends on the values of $N$ and $p_0$.*

In Theorem 3, we show the necessity of exploration when our initial belief is slightly wrong, i.e., it is slightly biased toward the other end of the network (In general, this could be due to a wrong belief, prior test results, etc). We have formally proved that when the testing capacity is limited, exploration can significantly improve the cumulative infections, i.e., contain the spread. This motivates the design of exploration policies. Even though the setting is simple, the phenomena it captures is much more general.

## 5 Simulations

### 5.1 Overview

In this section, we use simulations to study the performance of the proposed exploitation and exploration policies for various synthetic and real-data networks. Towards this end, we define some metrics that quantify how different metrics perform and key network parameters and attributes that determine the values of these metrics and thereby how exploitation and exploration compare. We also identify benchmark policies which represent the extreme ends of the tradeoff between exploration and exploitation to compare with the policies we propose and assess the performance enhancements brought about by judicious combinations of exploration and exploitation. Through our experiments, we aim to answer two main questions for various synthetic and real-data networks: (i) Can exploration policies do better that exploitation policies and if so, when would that be the case? (ii) What parameters would affect the performance of exploration and exploitation policies? These are important questions to shed light on the role of exploration. These questions are particularly raised by Theorem 3 in which we prove that exploration can significantly outperform exploitation in some (stylized) networks. We design the experiments in order to shed light on the above questions and to understand the extent of the necessity of exploration in different network models and scenarios.

**Network parameters** We consider the following parameters: (i) The unregulated delay $\ell$ which is the time from the initial start of the spread to the first time testing and intervention starts; (ii) The *(global) clustering coefficient* [61, Chapter 3], denoted by $\gamma_c$, which is defined as a measure of the degree to which nodes in a graph tend to cluster together; (iii) The *path-length*, denoted by $L_p$, which measures the average

shortest distance between every possible pair of nodes. We consider attributes such as the initialization of the process, and the lack of knowledge about $\{\underline{v}_i(t)\}_i$.

**Performance metrics** We consider the expected number of infected nodes in a time horizon $[0, T]$ as the performance measure for various policies. Let $C^0(T)$ be the number of infected nodes if there is no testing and isolation, $C^{RbEx}(T), C^{REEr}(T)$ be the corresponding numbers respectively for the *RbEx* policy (Algorithm 2) and the *REEr* policy (Algorithm 3). We consider a ratio between the expectations of these:

$$\texttt{Ratio} = \frac{\mathbb{E}[C^{RbEx}(T)] - \mathbb{E}[C^{REEr}(T)]}{\mathbb{E}[C^0(T)]}. \tag{18}$$

We define the *estimation error* $\texttt{Err}_\pi(t)$ towards capturing the impact of the lack of knowledge about $\{\underline{v}_i(t)\}_i$.

$$\texttt{Err}_\pi(t) = \frac{1}{N(t)} \sum_{i \in \mathcal{G}(t)} ||\underline{v}_i(t) - \underline{u}_i(t)||_2^2. \tag{19}$$

We consider the difference between the estimation errors of *RbEx* and *REEr* policies: $\Delta_{\texttt{Err}} = \texttt{Err}_{RbEx}(T) - \texttt{Err}_{REEr}(T)$.

**Benchmark policies** We will compare the proposed policies with 4 benchmark policies. (i) (Forward) Contact Tracing: we tested every day the nodes who have infectious neighbors (in a forward manner), denoted by *candidate nodes*. Only some candidate nodes are selected randomly due to testing resources being limited. Note that only exploitation is utilized under this benchmark. (ii) Random Testing: Every day, we randomly select nodes to test. Typical testing policies that could come out of SIR optimal control formulations for our problem would naturally reduce to random testing as they treat all nodes to be statistically identical and ignore the impact of network topology. One can interpret that random testing implements exploration to its full extent. (iii) Contact Tracing with Active Case Finding: A small portion of (for example, 5%) testing budget is utilized for active case finding [38]. This portion of the testing budget is used to test nodes by Random Testing. The remaining budget is utilized for forward contact tracing. (iv) Logistic Regression: We use ideas presented in [51], where simple classifiers were proposed based on the features of real data. In our setting, we choose the classifier to be based on logistic regression, and we define the feature of node $i$ as $X_i(t) = [1, n_i(t) + \epsilon]^T$. Here, $n_i(t)$ is the number of quarantined neighbors node $i$ has contacted before and including day $t$, and $\epsilon \neq 0$ is a superparameter aiming to avoid the case where $n_i(t) = 0$. In simulations, we set $\epsilon = 0.1$. Let the observation $Y_i(t)$ be the testing result of node $i$. In particular, if node $i$ is not tested on day $t$, then we do *not* collect the data $(X_i(t), Y_i(t))$. Thus, the probability of node $i$ being infectious is defined as the *Sigmoid* function

$$\frac{1}{1 + \exp(-X_i(t) \cdot w^T)},$$

where $w$ is the parameter which should be learned.

**Simulation Setting** We consider a process as described in Section 2 with $n_0$ randomly located initial infected nodes. The process evolved without any testing/intervention for $\ell$ days and we refer to $\ell$ as the *unregulated delay*. After that, one of the (initial) infectious nodes, denoted by node $i_0$, is (randomly) provided to the policies. Subsequently, the initial estimated probability vector is set to $\underline{u}_{i_0}(\ell) = (1, 0, 0, 0)$, and $\underline{u}_i(\ell) = (0, 0, 0, 1)$ when $i \neq i_0$. We consider the budget to be equal to the expected number of infected nodes at time $t$, i.e., $B(t) = \sum_{j=1}^{N(t)} v_I^{(j)}(t)$.

We choose model parameters considering the particular application of COVID-19 spread. In particular, 1) the mean latency period is $1/\lambda = 1$ or 2 days [56]; 2) the mean duration in the infectious state (I) is $1/\gamma = 7 \sim 14$ days [56, 57, 58]; 3) we choose the transmission rate $\beta$ in a specific network such that after a long time horizon, if no testing and isolation policies were applied, then around $60 \sim 90$ percent individuals are infected. We did not consider the case where 100 percent individuals are infected because given the recovery rate (and the topology), the spread may not reach every node.

We consider both synthetic networks such as Watts-Strogatz (WS) networks [62], Scale-free (SF) networks [63], Stochastic Block Models (SBM) [64] and a variant of it (V-SBM), as well as real-data networks. Descriptions and further results for the synthetic networks and real networks are presented in Appendix M.

**Watts-Strogatz Networks.** We consider a network $\text{WS}(N, d, \delta)$ with $N$ nodes, degree $d$, and rewiring probability $\delta$. The transmission probability of the spread is set to $\beta = 0.4$ and the number of initial seed is $n_0 = 3$.

**Scale-free Networks.** We consider a network $\text{SF}(N, \alpha)$ with $N$ nodes, and the fraction of nodes with degree $k$ follows a power law $k^{-\alpha}$, where $\alpha = 2.1, 2.3, 2.5, 2.7, 2.9$. The transmission probability of the spread is set to $\beta = 0.5$ and the number of initial seeds is $n_0 = 3$.

**Stochastic Block Models.** The SBM is a generative model for random graphs. The graph is divided into several communities, and subsets of nodes are characterized by being connected with particular edge densities. The intra-connection probability is $p_1$, and the inter-connection probability is $p_2$. We denote the SBM as $\text{SBM}(N, M, p_1, p_2)^7$. The transmission probability of the spread is set to $\beta = 0.04$ and the number of initial seed is $n_0 = 3$. The construction of SBM is given in Appendix M.1.

**A Variant of Stochastic Block Models.** Different from SBM, we only allow nodes in cluster $i$ to connect to nodes in successive clusters (the neighbor clusters). Denote a variant of SBM as $\text{V-SBM}(N, M, p_1, p_2)$. The transmission probability of the spread is set to $\beta = 0.04$ and the number of initial seed is $n_0 = 3$. The construction of V-SBM is given in Appendix M.1.

**Real-data Network I.** We consider a contact network of university students in the Copenhagen Networks Study [65]. The network is built based on the proximity between participating students recorded by smartphones, at 5 minute resolution. According to the definition of close contact by [58], we only used proximity events between individuals that lasted more than 15 minutes to construct the daily contact network. The contact network has 672 individuals spanning 28 days. To guarantee a long time-horizon, we replicate the contact network 4 times so that the time-horizon is 112 days. We set $\beta = 0.05$ and $n_0 = 5$ to have a realistic simulation of the Covid-19 spread. Note that the network is relatively dense, so we choose a relatively small value of $\beta$ to avoid the unrealistic case in which the disease spreads very fast (see Figure 11 (left)).

**Real-data Network II.** We consider a publicly available dataset on human social interactions collected specifically for modeling infectious disease dynamics [66, 67, 68]. The data set consists of pairwise distances between users of the BBC Pandemic Haslemere app over time. The contact network has 469 individuals spanning 576 days. Since the network is very sparse, then we compress contacts among individuals during 4 successive days to one day. Then, we have 469 individuals spanning 144 days. We set $\beta = 0.95$ and $n_0 = 30$ to have a realistic simulation of the Covid-19 spread. Note that the network is relatively sparse, so we choose a relatively large value of $\beta$ to avoid the unrealistic case in which the disease spreads very slow (see Figure 11 (left)).

## 5.2 Simulation Results in Synthetic networks

In this section, we compare the performances of our proposed policies and the benchmarks (defined in Section 5.1) in synthetic networks. We start with some specific networks and parameters for this purpose (see Figure 5, Figure 6, Figure 7, and Figure 8). The figures reveal that our proposed policies, i.e., the RbEx and REEr policies, outperform the benchmarks. In particular, in Figure 5 and Figure 6 (i.e., the WS and SF networks), the REEr policy outperforms the RbEx policy, and the REEr policy provides a more accurate estimation for $\{\underline{v}_i(t)\}_i$. In Figure 7 and Figure 8 (i.e., the SBM and V-SBM networks), the RbEx policy outperforms the REEr policy, and the RbEx policy provides a more accurate estimation for $\{\underline{v}_i(t)\}_i$. In addition, in Figure 5, we show that Algorithm 1 outperforms the RbEx policy but performs worse than the REEr policy (recall that the compuation time of Algorithm 1 is high, we therefore only plot the performance of Algorithm 1 in Figure 5 as an example). This implies that without exploration, the exploitation in a greedy manner can not perform well in WS networks.

From the discussions above, the advantages of exploration in distinct settings (different network topologies with variant parameters) are different. To investigate the advantages of exploration in distinct settings, it suffices to show how the main parameters affect the exploration. In this work, we consider three main parameters which are defined in Section 5.1, i.e., the unregulated delay $\ell$, the global clustering coefficient $\gamma_c$, and the path-length $L_p$. Detailed discussions are later given in Section 5.2.1.

---

$^7$Here, we assume that $M$ is an exact divisor of $N$.

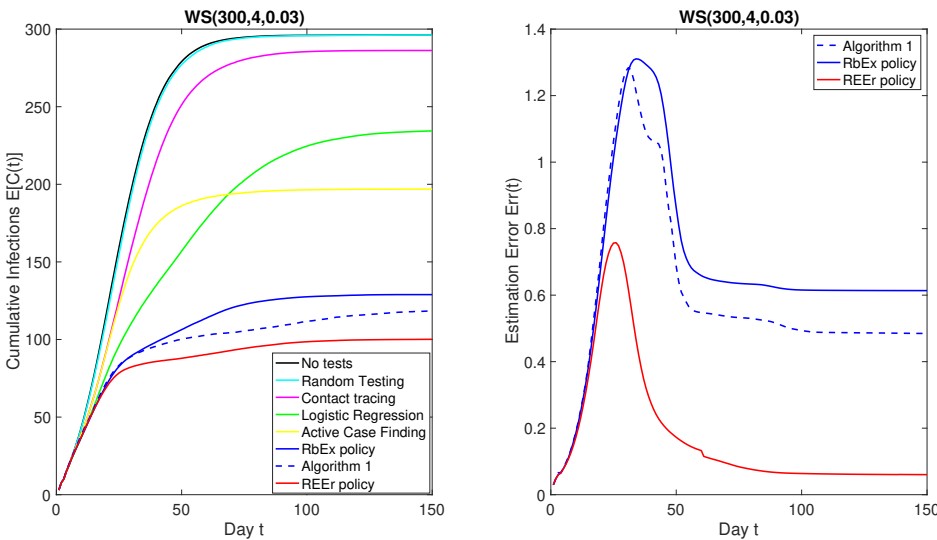

Figure 5: Performances and estimation errors of different policies in WS(300, 4, 0.03) when $\ell = 3$.

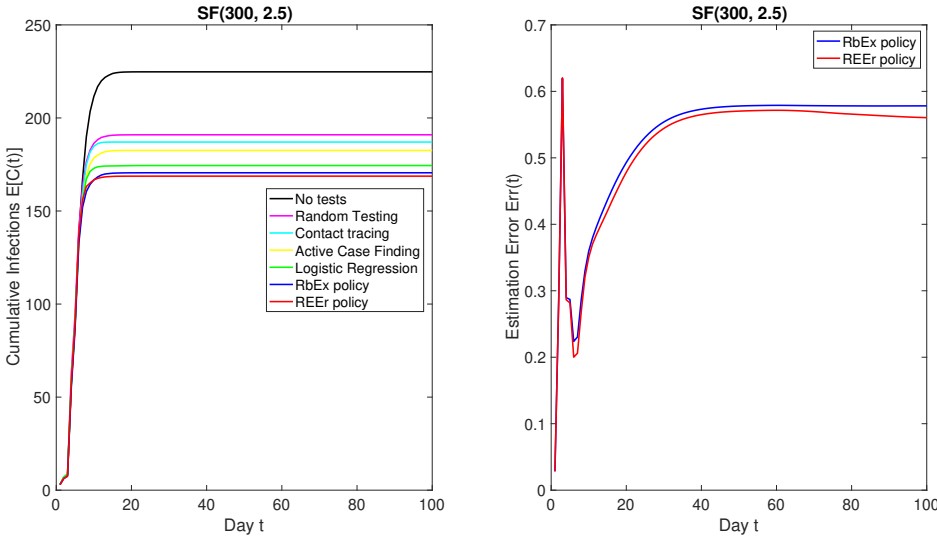

Figure 6: Performances and estimation errors of different policies in SF(300, 2.5) when $\ell = 3$.

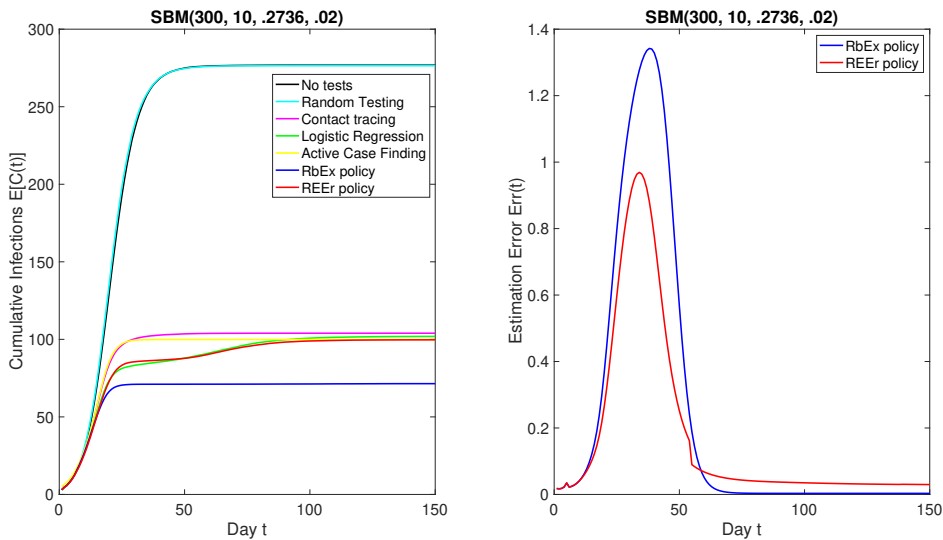

Figure 7: Performances and estimation errors of different policies in SBM(300, 10, .2736, .02) when $\ell = 5$.

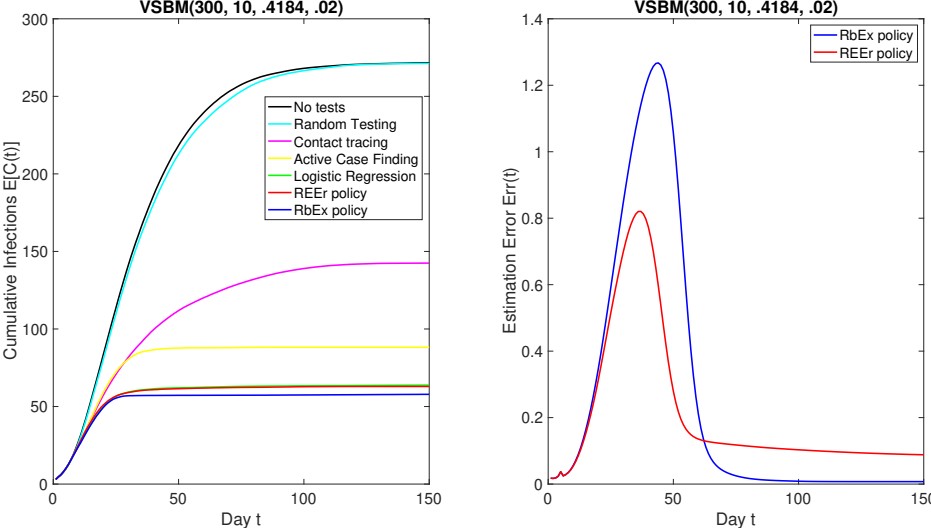

Figure 8: Performances and estimation errors of different policies in VSBM(300, 10, .4184, .02) when $\ell = 5$.

### 5.2.1 Impact of Network Parameters

In this subsection, we consider the impact of network parameters on the tradeoff between exploration and exploitation.

**Impact of $\ell$.** We first investigate the impact of the unregulated delay, $\ell$. Specifically, from Table 2, Table 3, Table 4, and Table 5, as $\ell$ increases, so does `Ratio` and $\Delta_{\texttt{Err}}$, implying that exploration becomes more effective. With increase in $\ell$, the infection continues in the network for longer, there are greater number of infectious nodes in the network and they are scattered throughout the network, thus exploration is better suited to locate them. Thus, the REEr policy can contain the spread of the disease faster.

In particular, the REEr policy is always better in WS networks. This is because exploitation may confine the tests in neighborhoods of some infected nodes. While in the SBM networks, the RbEx policy always outperforms the REEr policy. In both the SF and V-SBM networks, the RbEx policy is better when $\ell$ is small, and the REEr policy is better when $\ell$ is large. One interesting observation is that in the V-SBM networks, the REEr policy performs better when $\ell$ is large $(= 11, 13)$, but the corresponding estimation errors are larger than those in the RbEx policy. In this specific network topology, it appears that smaller estimation error does not always correspond to better cumulative infections. One potential reason is that the REEr policy is sensitive to $\ell$ in this topology, i.e., we can achieve smaller cumulative infections under the REEr policy even if the estimation error is larger.

| WS, $\ell$ | 3 | 5 | 7 | 9 | 11 |
|---|---|---|---|---|---|
| `Ratio` | 0.097 | 0.128 | 0.177 | 0.207 | 0.297 |
| $\Delta_{\texttt{Err}}$ | 0.553 | 0.814 | 1.092 | 1.197 | 1.449 |

Table 2: Role of the unregulated delay $\ell$ when $\delta = 0.03$.

| SF, $\ell$ | 3 | 5 | 7 | 9 | 11 |
|---|---|---|---|---|---|
| `Ratio` | $-0.0009$ | 0.0026 | 0.0033 | 0.0042 | 0.0059 |
| $\Delta_{\texttt{Err}}$ | $-0.0014$ | 0.0237 | 0.0334 | 0.0434 | 0.1212 |

Table 3: Role of the unregulated delay $\ell$ when $\alpha = 2.1$.

| SBM, $\ell$ | 5 | 7 | 9 | 11 | 13 |
|---|---|---|---|---|---|
| `Ratio` | $-0.092$ | $-0.079$ | $-0.042$ | $-0.035$ | $-0.025$ |
| $\Delta_{\texttt{Err}}$ | $-0.026$ | $-0.015$ | $-0.010$ | $-0.009$ | $-0.009$ |

Table 4: Role of the unregulated delay $\ell$ when $(p_1, p_2) = (.274, .02)$.

| V-SBM, $\ell$ | 5 | 7 | 9 | 11 | 13 |
|---|---|---|---|---|---|
| `Ratio` | $-0.022$ | $-0.016$ | $-0.007$ | 0.011 | 0.019 |
| $\Delta_{\texttt{Err}}$ | $-0.081$ | $-0.066$ | $-0.046$ | $-0.033$ | $-0.025$ |

Table 5: Role of the unregulated delay $\ell$ when $(p_1, p_2) = (.418, .02)$.

**Impact of $\gamma_c$ and $L_p$.** Then, we investigate the impact of the global clustering coefficient, i.e., $\gamma_c$, and the average shortest path-length, i.e., $L_p$. In Table 6, both $\gamma_c$ and $L_p$ decrease as $\delta$ increases. In Table 7, $\gamma_c$ decreases as $\alpha$ increases. For the SF networks, the graphs are often disconnected, so we only calculate $\gamma_c$ in Table 7. In Table 8 and Table 9, both $\gamma_c$ and $L_p$ decrease as $p_2$ increases.

From these tables, as $L_p$ or $\gamma_c$ decreases, the benefits of exploration compared to exploitation decrease as well. This confirms the intuition that exploration is particularly helpful in clustered networks with larger path lengths where undetected infection can spread without any intervention as exploitation largely confines the tests in neighborhoods of the infections that were previously detected. This is also supported by the fact that exploration lowers estimation error in such scenarios, as shown in Table 6, Table 7, Table 8, and Table 9. Furthermore, we investigate the role of $\gamma_c$ and $L_p$ individually in Appendix M.2.

| WS, $\delta$ | $\gamma_c$ | $L_p$ | Ratio | $\Delta_{\texttt{Err}}$ |
|---|---|---|---|---|
| 0 | .5 | 62.876 | 0.191 | 1.153 |
| .0075 | .489 | 21.264 | 0.182 | 1.423 |
| .015 | .473 | 14.253 | 0.174 | 0.991 |
| .0225 | .467 | 12.171 | 0.126 | 0.779 |
| .03 | .456 | 10.81 | 0.097 | 0.554 |

Table 6: Role of clustering coefficient and path length when $\ell = 3$.

| SF, $\alpha$ | $\gamma_c$ | Ratio | $\Delta_{\texttt{Err}}$ |
|---|---|---|---|
| 2.1 | .5017 | 0.0080 | 0.0334 |
| 2.3 | .3374 | 0.0057 | 0.0253 |
| 2.5 | .2348 | 0.0032 | 0.0177 |
| 2.7 | .1496 | $-0.0019$ | 0.0124 |
| 2.9 | .0219 | $-0.0064$ | 0.0081 |

Table 7: Role of clustering coefficient and path length $\ell = 3$.

| SBM, $(p_1, p_2)$ | $\gamma_c$ | $L_p$ | Ratio | $\Delta_{\texttt{Err}}$ |
|---|---|---|---|---|
| $(0.274, 0.02)$ | 0.111 | 2.573 | $-0.092$ | $-0.026$ |
| $(0.214, 0.026)$ | 0.075 | 2.518 | $-0.103$ | $-0.023$ |
| $(0.159, 0.032)$ | 0.056 | 2.492 | $-0.113$ | $-0.026$ |
| $(0.102, 0.039)$ | 0.048 | 2.480 | $-0.118$ | $-0.023$ |
| $(0.045, 0.045)$ | 0.043 | 2.455 | $-0.124$ | $-0.027$ |

Table 8: Role of clustering coefficient and path length $\ell = 5$.

| V-SBM, $(p_1, p_2)$ | $\gamma_c$ | $L_p$ | Ratio | $\Delta_{\texttt{Err}}$ |
|---|---|---|---|---|
| $(0.418, 0.020)$ | 0.3557 | 4.4264 | $-0.022$ | $-0.081$ |
| $(0.351, 0.052)$ | 0.2365 | 3.6584 | $-0.091$ | $-0.045$ |
| $(0.284, 0.085)$ | 0.1769 | 3.307 | $-0.104$ | $-0.055$ |
| $(0.217, 0.085)$ | 0.1385 | 3.1562 | $-0.112$ | $-0.041$ |
| $(0.150, 0.0150)$ | 0.1170 | 3.0563 | $-0.123$ | $-0.042$ |

Table 9: Role of clustering coefficient and path length $\ell = 5$.

| Real-data Network I, $\ell$ | 5 | 8 | 11 |
|---|---|---|---|
| Ratio | $-0.0559$ | $-0.0255$ | 0.009 |
| $\Delta_{\texttt{Err}}$ | $-0.061$ | $-0.030$ | 0.035 |

Table 10: Role of the unregulated delay $\ell$

| Real-data Network II, $\ell$ | 5 | 8 | 11 |
|---|---|---|---|
| Ratio | 0.0808 | 0.1039 | 0.1208 |
| $\Delta_{\texttt{Err}}$ | 0.0317 | 0.0535 | 0.0615 |

Table 11: Role of the unregulated delay $\ell$

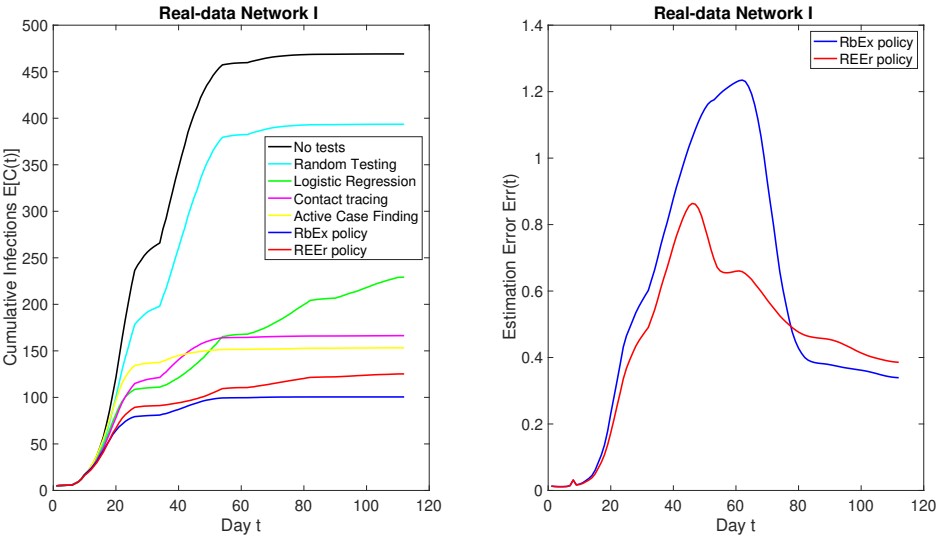

Figure 9: Performances and estimation errors of different policies in the real-data network I when $\ell = 8$.

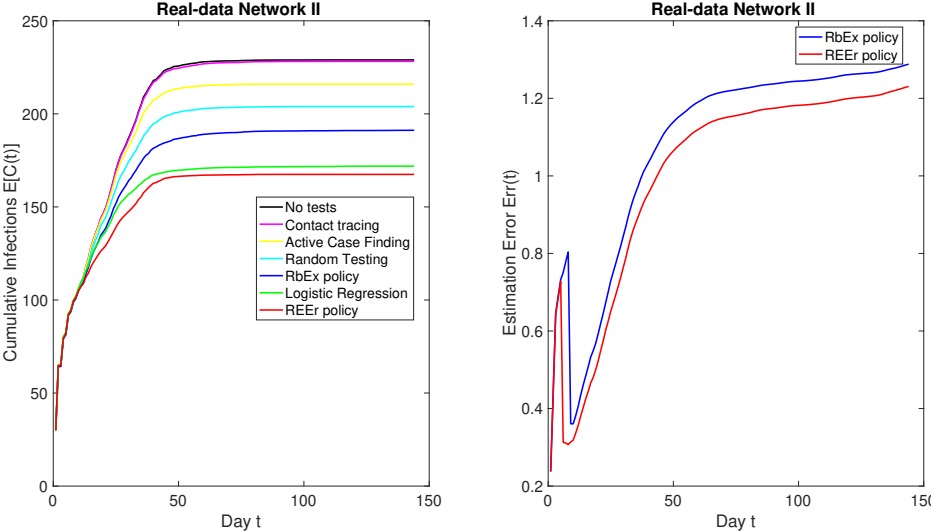

Figure 10: Performances and estimation errors of different policies in the real-data network II when $\ell = 8$.

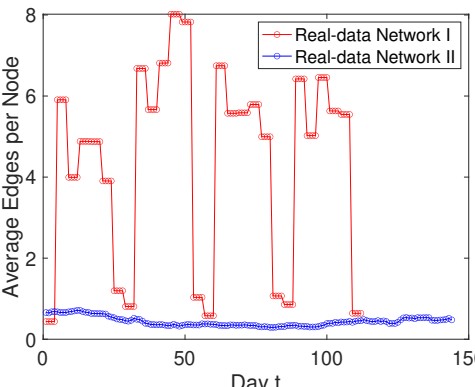 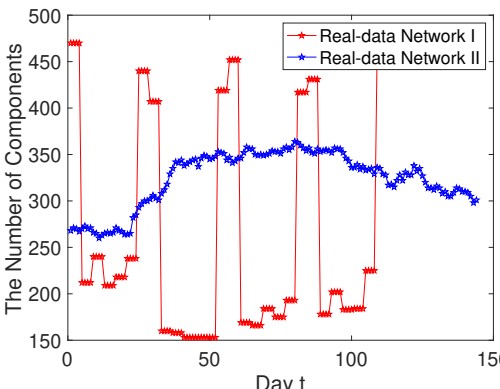

Figure 11: Left: The average number of edges per node on each day. Right: The number of components on each day.

### 5.3 Simulation Results in Real-data Networks

In this section, we verify our proposed policies in real data networks (Real-data Network I and Real-data Network II). In Figure 9, our proposed policies outperform the baselines, and the RbEx policy outperforms the RREr policy. In Figure 10, the REEr policy can contain the spread and outperform other baselines and RbEx, while the Logistic Regression policies outperforms RbEx. Comparing Figure 9 and Figure 10, we find that the RbEx policy performs well in Real-data Network I (better than the REEr policy), but performs not well in Real-data Network II (much worse than the REEr policy). In Figure 11 (left), we calculate the average edges per node on every day, and in Figure 11 (right), we calculate the number of components on every day. From Figure 11 (left), the Real-data Network I is denser than the Real-data Network II. However, from Figure 11 (right), the Real-data Network II often has more components (subgraphs) than the Real-data Network I. Thus, exploitation may become confined within some components (subgraphs), and fail to locate infectious nodes elsewhere, and exploration becomes more effective in presence of a large number of components. This explains the relative performances of REEr and RBEx in these. Contact tracing policy employs only exploitation, while active case finding policy uses most of its test budget for exploitation (and the small amount of the residual test budget for exploration). From Figure 9 and Figure 10, the contact tracing and the active case finding policies perform relatively poorly in the Real-data Network II compared to that in the Real-data Network I; this may again be attributed to the presence of a large number of components in the former.

As $\ell$ increases, as we show in Table 10 and Table 11 that the benefit of exploitation decreases. In Table 11, because of a large number of components, exploration always outperforms exploitation. However, in Table 10, we observe that exploration outperforms exploitation only for larger values of $\ell$. Our results are thus consistent with synthetic networks.

## 6 Conclusions and Future Work

In this paper, we studied the problem of containing a spread process (e.g. an infectious disease such as COVID-19) through sequential testing and isolation. We modeled the spread process by a compartmental model that evolves in time and stochastically spreads over a given contact network. Given a daily test budget, we aimed to minimize the cumulative infections. Under mild conditions, we proved that the problem can be cast as minimizing a supermodular function expressed in terms of nodes' probabilities of infection and proposed a greedy testing policy that attains a constant factor approximation ratio. We subsequently designed a computationally tractable reward-based policy that preferentially tests nodes that have higher rewards, where the reward of a node is defined as the expected number of new infections it induces in the next time slot. We showed that this policy effectively minimizes an upper bound on the cumulative infections.

These policies, however, need knowledge about nodes' infection probabilities which are unknown and evolving. Thus, they have to be actively learned by testing. We discussed how testing has a dual role in this problem:

(i) identifying the infected nodes and isolating them in order to contain the spread, and (ii) providing better estimates for the nodes' infection probabilities. We proved that this dual role of testing makes decision making more challenging. In particular, we showed that reward based policies that make decisions based on nodes' estimated infection probabilities can be arbitrarily sub-optimal while incorporating simple forms of exploration can boost their performance by a constant factor. Motivated by this finding, we devised exploration policies that probabilistically test nodes according to their rewards and numerically showed that when (i) the unregulated delay, (ii) the global clustering coefficient, or (iii) the average shortest path length increase, exploration becomes more beneficial as it provides better estimates of the nodes' probabilities of infection.

Given the history of observations, computing nodes' estimated probabilities of infection is itself a core challenge in our problem. We developed a message-passing framework to estimate these probabilities utilizing the observations in form of the test results. This framework passes messages back and forth in time to iteratively predict the probabilities in future and correct the errors in the estimates in prior time instants. This framework can also be of independent interest.

We showed novel tradeoffs between exploration and exploitation, different from the ones commonly observed in multi-armed bandit settings: (i) in our setting, the number of arms is time-variant and actions cannot be repeated; (ii) the tradeoffs in our setting are not due to lack of knowledge about the network or the process model, but rather due to lack of knowledge about the time-evolving unknown set of infected nodes.

We now describe directions for future research.

Our framework can be extended to incorporate delay and/or error in test results in a relatively straightforward manner (an outline of the extension incorporating a delay is given in Appendix I), but generalizing the performance guarantees for the proposed policies in these cases forms a direction of future research. This includes establishing fundamental lower bounds using genie-aided myopic policies.

## 6.1 Impact Statements

We have made several assumptions for the purpose of analytical and computational tractability which do not hold in practice: (1) the infections from different nodes are independent (2) given the entire history of testing results the states of nodes on the truncation day are independent (Assumption 1), (3) the symptoms need not be considered in deciding who should be tested and (4) the public health authority knows the entire network topology and uses it to determine who should be tested (5) independence of states of nodes (in one step). The first two assumptions were used to derive the message passing framework and to prove that the objective function is super-modular which in turn led to a myopic testing strategy which is also optimal. The first assumption is reasonable as specific actions of infected individuals, eg, coughing, touching, spread the infection, which are undertaken independently.

We now consider the second assumption, ie, Assumption 1, in which we assume that the nodes' states $\zeta(t-g)$ (in the posterior probability space on day $t-g$) are independent. Note that $g$ is the truncation time for each backward step, that is, once we get the observations $\underline{Y}(t)$, we do the backward step and truncate at time $t-g$. This assumption does not impose independence on the state of the nodes, but only in the posterior space at a specific time. That is, in the process of propagating information back to time $t-g$, we are assuming that there is no further correlation between time $t-g-1$ and time $t-g$ worthwhile to exploit given observations at time $t$. Naturally, as $g$ gets larger and larger, our framework and calculations become more precise, as the impact of the testing results at time $t$ in inferring about the nodes' probabilities at time $t-g$ vanishes as $g$ gets large. But increase in $g$ significantly increases the computation time. Therefore, for computational tractability, of the backward update equations, we use $g=1$. In principle the derivations of the backward update equations can be generalized in a straightforward manner to $g>1$. But designing approximation strategies that ensure computational tractability for larger $g$ constitutes a direction of future research.

Consider the third assumption. We have not considered symptoms in determining who to test. But for some infectious diseases, symptoms are a reliable manifestation of the disease (e.g., Ebola). In principle our testing framework can be generalized in a straightforward manner to consider symptoms by introducing additional states in the compartmental model for evolution of the disease. But introduction of additional

states significantly increases the computation time, for example of the forward and backward updates of the probabilities that individuals have the disease, which renders implementation of our framework challenging. Considering symptoms while retaining computational tractability constitutes a direction of future research.

Next, consider the fourth assumption. In practice, public health authorities will not typically know contact networks in their entirety particularly when they are large, for example, as in large cities. However, small network topologies, for example, contact networks within a community, may be observed by the public health authority. As a specific example, the Government of China fully detected contact networks in many communities in Wuhan and tracked paths traversed by every individual [69]. This tracking may also generate concerns about privacy which is beyond the scope of this paper. Nonetheless, the technology for learning contact networks in their entirety for small communities exists and our framework can be utilized for those. Generalizing our framework to obtain approximation guarantees when contact networks can only be partially observed constitutes a direction of future research.

Finally consider the last assumption. Note that it is a strong assumption and clearly does not hold in general but it has been resorted to for only one step in the entire framework. Specifically to obtain Equation (5) we have assumed that the state of the nodes are independent. This allows us to obtain a simple expression in (5) in terms of the infection probabilities. We do not use this independence assumption in the rest of the paper.

### Acknowledgments

This work was supported by NSF CAREER Award 2047482, NSF Award 1909186, NSF Award 1910594, and NSF Award 2008284.

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

## A   Proof of Lemma 1

Note that a node is counted in $C^\pi(t)$ once it has been infected. Then, on day $t + 1$, $C^\pi(t + 1)$ increases (comparing to $C^\pi(t)$) only because some susceptible nodes are infected by infectious nodes and are in the latent state for the first time.

After testing, positive nodes in $\mathcal{K}^\pi(t)$ would not infect others because they are quarantined, and negative nodes would not infect others due to the model assumptions. Hence

$$C^\pi(t + 1) = C^\pi(t) + \sum_{i \in \mathcal{V}(t)} F_i(\mathcal{V}(t) \backslash \mathcal{K}^\pi(t); t).$$

Taking the expectation on both sides, we obtain the desired result.

## B   Proof of Theorem 1

To show $S\big(\mathcal{K}^\pi(t); t\big)$ defined in (6) is a supermodular function. It suffices to show that for any $\mathcal{A} \subset \mathcal{B} \subset \mathcal{V}(t)$, and for $x \in \mathcal{V}(t) \backslash \mathcal{B}$, we have

$$S\big(\mathcal{A} \cup \{x\}; t\big) - S\big(\mathcal{A}; t\big) \leq S\big(\mathcal{B} \cup \{x\}; t\big) - S\big(\mathcal{B}; t\big). \tag{20}$$

Then, it suffices to show for any $i \in \mathcal{V}(t)$,

$$\mathbb{E}[F_i\big(\mathcal{A} \cup \{x\}; t\big)] - \mathbb{E}[F_i\big(\mathcal{A}; t\big)] \leq \mathbb{E}[F_i\big(\mathcal{B} \cup \{x\}; t\big)] - \mathbb{E}[F_i\big(\mathcal{B}; t\big)]. \tag{21}$$

Now, we consider three cases.

**Case 1**. If $\mathcal{A} \cap \partial_i(t) = \mathcal{B} \cap \partial_i(t)$, then from (5), the LHS and RHS in (21) are exactly the same. Hence, (21) holds.

**Case 2**. If $\mathcal{A} \cap \partial_i(t) \subset \mathcal{B} \cap \partial_i(t)$, and $x \notin \partial_i$, then from (5), $f_i(\mathcal{A} \cup \{x\}) = f_i(\mathcal{A})$ and $f_i(\mathcal{B} \cup \{x\}) = f_i(\mathcal{B})$. Hence (21) holds.

**Case 3**. If $\mathcal{A} \cap \partial_i(t) \subset \mathcal{B} \cap \partial_i(t)$, and $x \in \partial_i(t)$, let $\mathcal{Y} = \big(\mathcal{B} \cap \partial_i(t)\big) \backslash \big(\mathcal{A} \cap \partial_i(t)\big)$. Here $x \notin \mathcal{Y}$. From (5), we can compute

$$\mathbb{E}[F_i(\mathcal{A} \cup \{x\}; t)] - \mathbb{E}[F_i(\mathcal{A}; t)]$$
$$= v_S^{(i)}(t) \prod_{j \in \partial_i(t) \backslash (\mathcal{A} \cup \{x\})} \big(1 - \beta v_I^{(j)}(t)\big)$$
$$\times \Big(1 - \prod_{j \in \partial_i(t) \cap (\mathcal{A} \cup \{x\})} (1 - \beta v_I^{(j)}(t)) - (1 - \beta v_I^{(x)}(t))\big(1 - \prod_{j \in \partial_i(t) \cap \mathcal{A}} (1 - \beta v_I^{(j)}(t))\big)\Big),$$

which implies

$$
\begin{aligned}
&\mathbb{E}[F_i(\mathcal{A} \cup \{x\}; t)] - \mathbb{E}[F_i(\mathcal{A}; t)] \\
&= v_S^{(i)}(t) \prod_{j \in \partial_i(t) \backslash (\mathcal{A} \cup \{x\})} \big(1 - \beta v_I^{(j)}(t)\big) \\
&\quad \times \Big( \beta v_I^{(x)}(t) - \prod_{j \in \partial_i(t) \cap (\mathcal{A} \cup \{x\})} \big(1 - \beta v_I^{(j)}(t)\big) + \prod_{j \in \partial_i(t) \cap (\mathcal{A} \cup \{x\})} \big(1 - \beta v_I^{(j)}(t)\big) \big) \Big) \\
&= v_S^{(i)}(t) \prod_{j \in \partial_i(t) \backslash (\mathcal{A} \cup \{x\})} \big(1 - \beta v_I^{(j)}(t)\big) \beta v_I^{(x)}(t).
\end{aligned}
$$

Similarly, note that $\big(\mathcal{B} \cap \partial_i(t)\big) = \big(\mathcal{A} \cap \partial_i(t)\big) \cup \mathcal{Y}$. We have

$$
\begin{aligned}
&\mathbb{E}[F_i(\mathcal{B} \cup \{x\}; t)] - \mathbb{E}[F_i(\mathcal{B}; t)] \\
&= v_S^{(i)}(t) \prod_{j \in \partial_i(t) \backslash (\mathcal{A} \cup (\{x\} \cup \mathcal{Y}))} \big(1 - \beta v_I^{(j)}(t)\big) \beta v_I^{(x)}(t).
\end{aligned}
$$

Thus,

$$
\frac{\mathbb{E}[F_i(\mathcal{A} \cup \{x\}; t)] - \mathbb{E}[F_i(\mathcal{A}; t)]}{\mathbb{E}[F_i(\mathcal{B} \cup \{x\}; t)] - \mathbb{E}[F_i(\mathcal{B}; t)]} = \prod_{y \in \mathcal{Y}} \big(1 - \beta v_I^{(y)}(t)\big) \leq 1,
$$

which implies $S\big(\mathcal{TP}^\pi(t)\big)$ is supmodular.

To show $S\big(\mathcal{K}^\pi(t); t\big)$ is an increasing monotone function on $\mathcal{K}^\pi(t)$, it suffices to show $\mathbb{E}[F_i\big(\mathcal{K}^\pi(t); t\big)]$ is an increasing monotone function on $\mathcal{K}^\pi(t)$ for any $i$.

For $\mathcal{A} \subset \mathcal{B}$, we have $\partial_i(t) \backslash \mathcal{B} \subset \partial_i(t) \backslash \mathcal{A}$, and $\mathcal{A} \cap \partial_i(t) \subset \mathcal{B} \cap \partial_i(t)$. Then

$$
\prod_{j \in \partial_i(t) \backslash \mathcal{B}} \big(1 - \beta v_I^{(j)}(t)\big) \geq \prod_{j \in \partial_i(t) \backslash \mathcal{A}} \big(1 - \beta v_I^{(j)}(t)\big)
$$
$$
\prod_{j \in \mathcal{B} \cap \partial_i(t)} \big(1 - \beta v_I^{(j)}(t)\big) \leq \prod_{j \in \mathcal{A} \cap \partial_i(t)} \big(1 - \beta v_I^{(j)}(t)\big),
$$

and thus, from (5), we have $\mathbb{E}[F_i\big(\mathcal{A}; t\big)] \leq \mathbb{E}[F_i\big(\mathcal{B}; t\big)]$.

## C    Complexity of Algorithm 1

First of all, we consider the complexity of (5). Suppose $\{\underline{v}_i(t)\}_{i \in \mathcal{V}(t)}$ is given for every day $t$. For any $\mathcal{K}^\pi(t)$, the complexity of computing (5) is

$$
1 + |\partial_i(t) \backslash \mathcal{K}^\pi(t)| - 1 + 1 + |\partial_i(t) \backslash \mathcal{K}^\pi(t)| + |\partial_i(t) \cap \mathcal{K}^\pi(t)| - 1 + |\partial_i(t) \cap \mathcal{K}^\pi(t)| = 2|\partial_i(t)|.
$$

Then, for any $\mathcal{K}^\pi(t)$, the complexity of computing $S\big(\mathcal{K}^\pi(t); t\big)$ is

$$
2 \sum_{j \in \mathcal{V}(t)} |\partial_j(t)|.
$$

From Algorithm 1, in step $i$, the complexity is

$$
\big(N(t) - i + 1\big) \times 2 \sum_{j \in \mathcal{V}(t)} |\partial_j(t)|.
$$

And in total we have $\big(N(t) - |\mathcal{K}^\pi(t)|\big)$ steps, therefore, on day $t$ the complexity of Algorithm 1 is

$$
\sum_{i=0}^{N(t) - |\mathcal{K}^\pi(t)|} 2\big(N(t) - i + 1\big) \sum_{j \in \mathcal{V}(t)} |\partial_j(t)|.
$$

Recall that the time horizon is $T$, then the total complexity of Algorithm 1 is

$$\sum_{t=0}^{T-1} \sum_{i=0}^{N(t)-|\mathcal{K}^\pi(t)|} 2\big(N(t)-i+1\big) \sum_{j \in \mathcal{V}(t)} |\partial_j(t)|.$$

Note that

$$\sum_{i=1}^{N(t)-|\mathcal{K}^\pi(t)|} 2\big(N(t)-i+1\big) \leq O\big(N^2(t)\big)$$

$$\sum_{i \in \mathcal{V}(t)} |\partial_i(t)| \leq O\big(N^2(t)\big).$$

Then, the total complexity is bounded by

$$O\Big( \sum_{t=0}^{T-1} N^4(t) \Big).$$

## D    Proof of Lemma 2

As defined in [60] (an equivalent definition of footnote 3), consider a finite set $I$, $f : 2^I \to \mathbb{R}$ is a supermodular function if for all $X, Y \subset I$,

$$f(X \cup Y) + f(X \cap Y) \geq f(X) + f(Y). \tag{22}$$

Following the supermodularity of function $S(\cdot)$ as shown in Theorem 1, set $X = \mathcal{V}(t) \backslash \mathcal{K}^\pi(t)$ and $Y = \mathcal{K}^\pi(t)$ in (22), we have

$$S\big(\mathcal{V}(t) \backslash \mathcal{K}^\pi(t); t\big) \leq S\big(\mathcal{V}(t); t\big) - S\big(\mathcal{K}^\pi(t); t\big). \tag{23}$$

Again, set $X = \mathcal{K}^\pi(t) \backslash \{i\}$ and $Y = \{i\}$ in (22), and use (22) repeatedly to obtain:

$$S\big(\mathcal{K}^\pi(t); t\big) \geq \sum_{i \in \mathcal{K}^\pi(t)} S\big(\{i\}; t\big) = \sum_{i \in \mathcal{K}^\pi(t)} r_i(t). \tag{24}$$

Substituting (24) in (23), we obtain

$$S\big(\mathcal{V}(t) \backslash \mathcal{K}^\pi(t); t\big) \leq S\big(\mathcal{V}(t); t\big) - \sum_{i \in \mathcal{K}^\pi(t)} r_i(t).$$

## E    Local Transition Equations

In this section, we will describe the local transition matrix $\mathsf{P}_i\big(\{\underline{v}_j(t)\}_{j \in \partial_i^+(t)}\big)$ used in (12). The state of each node evolves as follows: (i) if node $i$ is susceptible on day $t$, then it might be infected by its neighbors in $\partial_i(t)$; (ii) an infectious node remains in the latent state with probability $1 - \lambda$, and changes state to the infectious state $(I)$ with probability $\lambda$; (iv) if node $i$ is in state $I$, it will recover after a geometric distribution with parameter $\gamma$. Let $\xi_i(t) = 1 - \prod_{m \in \partial_i(t)} \big(1 - v_I^{(m)}(t)\beta\big)$. In particular, define $\xi_i(t) = 0$ if $\partial_i(t) = \varnothing$. Then, the probabilities of nodes being in different states evolve in time as follows:

$$v_I^{(i)}(t+1) = v_I^{(i)}(t)(1-\gamma) + v_L^{(i)}(t)\lambda \tag{25}$$

$$v_L^{(i)}(t+1) = v_L^{(i)}(t)(1-\lambda) + v_S^{(i)}(t)\xi_i(t) \tag{26}$$

$$v_R^{(i)}(t+1) = v_R^{(i)}(t) + v_I^{(i)}(t)\gamma \tag{27}$$

$$v_S^{(i)}(t+1) = v_S^{(i)}(t)\big(1 - \xi_i(t)\big). \tag{28}$$

Note that row vector $\underline{v}_i(t)$ is defined in (3). Collecting (25) - (28), we define the local transition probability matrix as given below:

$$\mathrm{P}_i\big(\{\underline{v}_j(t)\}_{j\in\partial_i^+(t)}\big) = \begin{bmatrix} (1-\gamma) & 0 & \gamma & 0 \\ \lambda & 1-\lambda & 0 & 0 \\ 0 & 0 & 1 & 0 \\ 0 & \xi_i(t) & 0 & 1-\xi_i(t) \end{bmatrix}. \tag{29}$$

and we obtain (12).

## F  Proofs of (13) and (14)

First of all, we give the following definition.

**Definition 1.** *Let $X$ be a random variable and $\mathcal{B}$ be an event. Define $X|_{\mathcal{B}}$ as the random variable $X$ given $\mathcal{B}$; i.e.,*

$$\Pr\big(X|_{\mathcal{B}} = x\big) = \Pr\big(X = x|\mathcal{B}\big). \tag{30}$$

For brevity, let us define

$$\theta_i(t) = \sigma_i(t)|_{\{\underline{Y}(\tau)\}_{\tau=1}^{t-1}}, \quad \zeta_i(t) = \sigma_i(t)|_{\{\underline{Y}(\tau)\}_{\tau=1}^{t}}.$$

We thus have

$$u_x^{(i)}(t) = \Pr\big(\theta_i(t) = x\big), \quad w_x^{(i)}(t) = \Pr\big(\zeta_i(t) = x\big).$$

Recall that

$$\underline{v}_i(t) = \big[v_x^{(i)}(t)\big]_{x\in\mathcal{X}}, \ v_x^{(i)}(t) = \Pr\big(\sigma_i(t) = x\big).$$

Then, (12) can be re-written as

$$\Pr\big(\sigma_i(t+1) = x_i'\big) = \Pr\big(\sigma_i(t) = x_i\big)\mathrm{P}_i\Big(\{\sigma_j(t)\}_{j\in\partial_i^+(t)} = \{x_j\}_{j\in\partial_i^+(t)}\Big), \tag{31}$$

where $x_i', \{x_j\}_{j\in\partial_i^+(t)} \in \mathcal{X}$. Conditioning both sides of (31) on $\{\underline{Y}(\tau)\}_{\tau=1}^{t-1}$, state variables $\sigma_i(t)$ and $\sigma_i(t-1)$ in (31) can be replaced by $\theta_i(t)$ and $\zeta_i(t-1)$, respectively, to obtain

$$\underline{u}_i(t) = \underline{w}_i(t-1) \times \mathrm{P}_i\big(\{\underline{w}_j(t-1)\}_{j\in\partial_i^+(t-1)}\big), \tag{32}$$

which gives (13). In addition, define

$$\phi_i(t) = \sigma_i(t)|_{\{\underline{Y}(\tau)\}_{\tau=1}^{t+1}},$$

and

$$\begin{aligned} \underline{e}_i(t-1) &= (e_x^{(i)}(t-1), x \in \mathcal{X}), \\ e_x^{(i)}(t-1) &= \Pr\big(\phi_i(t-1) = x\big). \end{aligned} \tag{33}$$

This notation implies

$$\phi_i(t-1) = \theta_i(t-1)|_{\underline{Y}(t)}. \tag{34}$$

Similarly, conditioning both sides of (32) on $\underline{Y}(t)$, we find

$$\underline{w}_i(t) = \underline{e}_i(t-1) \times \tilde{\mathrm{P}}_i\big(\{\underline{e}_j(t-1)\}_{j\in\partial_i^+(t-1)}\big), \tag{35}$$

which gives (14). $\tilde{P}_i(\{\underline{e}_j(t-1)\}_{j\in\partial_i^+(t-1)})$ is obtained in the following subsection.

### F.1 Computing the transition probability matrix $\tilde{P}_i(\{\underline{e}_j(t-1)\}_{j\in\partial_i^+(t-1)})$

Note that $\tilde{P}_i(\{\underline{e}_j(t-1)\}_{j\in\partial_i^+(t-1)})$ is not the same as $P_i(\{\underline{w}_j(t)\}_{j\in\partial_i^+(t)})$. *This is because "future" observations were available in* $\tilde{P}_i(\{\underline{e}_j(t-1)\}_{j\in\partial_i^+(t-1)})$. To get $\tilde{P}_i(\{\underline{e}_j(t-1)\}_{j\in\partial_i^+(t-1)})$, we split the nodes $\mathcal{V}(t)$ into two classes of nodes: (i) nodes who do not get new observations and (ii) nodes who get new observations. $\tilde{P}_i(\{\underline{e}_j(t-1)\}_{j\in\partial_i^+(t-1)})$ is obtained by the following rules. For the first class of nodes, the local transition matrix in (35), i.e., $\tilde{P}_i(\{\underline{e}_j(t-1)\}_{j\in\partial_i^+(t-1)})$, is the same as that in (32). However, for the second class of nodes, the local transition matrices are changed accordingly because of the new observations. Let $[A]_{\{i,:\}}$ be the $i^{th}$ row of matrix $A$, and $q_i$ be a $1\times 4$ vector with the $i^{th}$ element being one and the rest zero. For brevity, denote the local transition matrices in (32) and (35) by $\mathsf{P}_i(t-1)$ and $\tilde{\mathsf{P}}_i(t-1)$, respectively. We have the following three cases:

(i) If node $i$ is not observed, then node $i$ does not have new observation and we have

$$\tilde{P}_i(t-1) = P_i(t-1). \tag{36}$$

(ii) If $Y_i(t) = 0$, then node $i$ is not infectious in day $t$ with probability 1. The local transition matrix is changed to

$$[\tilde{\mathsf{P}}_i(t-1)]_{\{j,:\}} = \begin{cases} q_3 & j=1 \\ q_2 & j=2 \\ [\mathsf{P}_i(t-1)]_{\{j,:\}} & \text{otherwise} \end{cases}. \tag{37}$$

(iii) If $Y_i(t) = 1$, then node $i$ is infectious in day $t$ with probability 1. The local transition matrix is changed to

$$[\tilde{\mathsf{P}}_i(t-1)]_{\{j,:\}} = \begin{cases} q_1 & j=1 \\ q_1 & j=2 \\ [\mathsf{P}_i(t-1)]_{\{j,:\}} & \text{otherwise} \end{cases}. \tag{38}$$

## G Proofs of (16) and (17)

Using new observations, we aim to move backward in time and update our belief (posterior probability) in previous time slots. Define a *truncation number $g$* and suppose that $\{\underline{Y}(t)\}$ affects the posterior probabilities from day $t$ to day $t-g$. We call day $t-g$ the *truncation day* associated with day $t$. To get accurate posterior probabilities in every day, we need to set $g = t$ on every day $t$ and track back to the initial time. However, the influence weakens as time elapses backwards, and for computation tractability, we continue under the following assumption where $g = 1$. Recall that $\zeta_i(t) = \sigma_i(t)|_{\{\underline{Y}(\tau)\}_{\tau=1}^t}$.

**Assumption 1.** *On the truncation day $(t-g)$, $\{\zeta_i(t-g)\}_i$ are independent over $i$. In the following, the truncation number is assumed to be $g = 1$.*

**Remark 3.** *In Assumption 1, we assume that the nodes' states $\zeta(t-g)$ (in the posterior probability space on day $t-g$) are independent. This assumption is only used at time $t$ of our probability update in a moving window kind of way. It provides us with a truncation time for each backward step. In particular, under Assumption 1, once we get the observations $\underline{Y}(t)$, we do the backward step and truncate at time $t-g$. For example, in the trivial case of $g = t$, the assumption holds. This assumption does not impose independence on the state of the nodes, but only in the posterior space at a specific time. In a sense, in the process of propagating information back to time $t-g$, we are assuming that there is no further correlation between time $t-g-1$ and time $t-g$ worthwhile to exploit given observations at time $t$. Naturally, as $g$ gets larger and larger, our framework and calculations become more precise but this comes at a huge computational cost. The idea behind truncating the backward step lies in the observation that the impact of the testing results at time $t$ in inferring about the nodes' probabilities at time $t-g$ vanishes as $g$ gets large. For simplicity of derivations and to have manageable complexity, we set $g = 1$. The idea and the derivations can be generalized in a straightforward manner to larger $g$.*

Note that the posterior probabilities on day $t - 1$, $\underline{w}_i(t - 1)$, $i \in \mathcal{V}(t - 1)$, are assumed known (and are conditioned on the history of observations $\{\underline{Y}(\tau)\}_{\tau=1}^{t-1}$). The probability vector $\underline{e}_i(t - 1)$ is the new posterior probability at time $t - 1$ which is updated (from $\underline{w}_i(t-1)$) based on new observations $\underline{Y}(t)$. In other words, we infer about the previous state of the nodes given new observations at present time.

To obtain $\{\underline{w}_i(t)\}_{i \in \mathcal{V}(t)}$, it suffices to obtain $\underline{e}_i(t-1)$ and the corresponding local transition matrix $\tilde{P}_i(\{\underline{e}_j(t-1)\}_{j \in \partial_i^+(t-1)})$, see (14). Note that the posterior probabilities $\underline{w}_i(t - 1)$, $i \in \mathcal{V}(t - 1)$, which are calculated based on $\underline{Y}(t - 1)$, are known. The vector $\underline{e}_i(t - 1)$ is the new posterior probability which is updated based on $\underline{Y}(t)$ and $\underline{w}_i(t - 1)$.

Equation (16), which we aim to prove, simply follows from Definition 1, (33)-(34), and Bayes rule:

$$e_x^{(i)}(t - 1) = \Pr\left(\zeta_i(t - 1) = x | \underline{Y}(t)\right) = \frac{\Pr\left(\underline{Y}(t)|\zeta_i(t - 1) = x\right) w_x^{(i)}(t - 1)}{\Pr\left(\underline{Y}(t)\right)}. \tag{39}$$

To find $\Pr\left(\underline{Y}(t)|\zeta_i(t - 1) = x\right)$, and establish (17), we now proceed as follows. We introduce $\{\theta_j(t)\}_{j \in \mathcal{O}(t)}$ into (39). In particular, we have

$$\Pr\left(\underline{Y}(t)|\zeta_i(t - 1) = x\right) = \sum_{\theta_j(t), \ j \in \mathcal{O}(t)} \Pr\left(\{\theta_j(t)\}_{j \in \mathcal{O}(t)}, \underline{Y}(t)|\zeta_i(t - 1) = x\right)$$

By the chain rule of conditional probability,

$$\begin{aligned} &\Pr\left(\underline{Y}(t)|\zeta_i(t - 1) = x\right) \\ &= \sum_{\theta_j(t), \ j \in \mathcal{O}(t)} \Pr\left(\underline{Y}(t)|\{\theta_j(t)\}_{j \in \mathcal{O}(t)}, \zeta_i(t - 1) = x\right) \times \Pr\left(\{\theta_j(t)\}_{j \in \mathcal{O}(t)}|\zeta_i(t - 1) = x\right). \end{aligned}$$

From (15), $\{\zeta_j(t)\}_{j \in \mathcal{V}(t)}$ and $\{\theta_j(t)\}_{j \in \mathcal{V}(t)}$ are variables defined by $\{\sigma_j(t)\}_{j \in \mathcal{V}(t)}$ in posterior spaces of $\{\underline{Y}(\tau)\}_{\tau=1}^{t}$ and $\{\underline{Y}(\tau)\}_{\tau=1}^{t-1}$, respectively. Since $\underline{Y}(t)$ is a deterministic function of $\{\sigma_j(t)\}_{j \in \mathcal{O}(t)}$, and hence $\{\theta_j(t)\}_{j \in \mathcal{O}(t)}$, then $\underline{Y}(t)$ is independent of $\zeta_i(t - 1)$ given $\{\theta_j(t)\}_{j \in \mathcal{O}(t)}$. In addition, the testing result $Y_j(t)$ (on day $t$) of node $j$ only depends on its state, i.e., given $\theta_j(t)$, the testing results are determined. Therefore, we have

$$\Pr\left(\underline{Y}(t)|\{\theta_j(t)\}_{j \in \mathcal{O}(t)}, \zeta_i(t - 1) = x\right) = \Pr\left(\underline{Y}(t)|\{\theta_j(t)\}_{j \in \mathcal{O}(t)}\right) = \prod_{j \in \mathcal{O}(t)} \Pr\left(Y_j(t)|\theta_j(t)\right).$$

The product above is an indicator which takes values on $\{0, 1\}$. We can thus re-write it as follows:

$$\Pr\left(\underline{Y}(t)|\{\theta_j(t)\}_{j \in \mathcal{O}(t)}, \zeta_i(t - 1) = x\right) \triangleq \delta(\{Y_j(t), \theta_j(t)\}_{j \in \mathcal{O}(t)}).$$

where

$$\delta(\{Y_j(t), \theta_j(t)\}_{j \in \mathcal{O}(t)}) = 1$$

if the pairs $\{Y_j(t), \theta_j(t)\}_{j \in \mathcal{O}(t)}$ are consistent, and

$$\delta(\{Y_j(t), \theta_j(t)\}_{j \in \mathcal{O}(t)}) = 0$$

otherwise.

Next, define

$$\Theta_i(t) = \{j | j \in \partial_k^+(t - 1), k \in \mathcal{O}(t)\} \backslash \{i\}$$

to represent the neighbors (in day $t - 1$) of nodes in $\mathcal{O}(t)$ excluding node $i$. Then,

$$\Pr\left(\{\theta_j(t)\}_{j \in \mathcal{O}(t)}|\zeta_i(t - 1) = x\right) = \sum_{\zeta_l(t-1), \ l \in \Theta_i(t)} \Pr\left(\{\theta_j(t)\}_{j \in \mathcal{O}(t)}, \{\zeta_l(t-1)\}_{l \in \Theta_i(t)}|\zeta_i(t - 1) = x\right). \tag{40}$$

By the chain rule of conditional probability,

$$
\Pr\left(\{\theta_j(t)\}_{j\in\mathcal{O}(t)}|\zeta_i(t-1)=x\right)
$$
$$
= \sum_{\zeta_l(t-1),\ l\in\Theta_i(t)} \Pr\left(\{\theta_j(t)\}_{j\in\mathcal{O}(t)}|\{\zeta_l(t-1)\}_{l\in\Theta_i(t)},\zeta_i(t-1)=x\right) \times \Pr\left(\{\zeta_l(t-1)\}_{l\in\Theta_i(t)}|\zeta_i(t-1)=x\right).
$$

Given $\{\zeta_l(t-1)\}_{l\in\Theta_i(t)}\cup\{\zeta_i(t-1)\}$, $\{\theta_j(t)\}_{j\in\mathcal{O}(t)}$ are independent. We thus have

$$
\Pr\left(\{\theta_j(t)\}_{j\in\mathcal{O}(t)}|\{\zeta_l(t-1)\}_{l\in\Theta_i(t)},\zeta_i(t-1)=x\right)
$$
$$
= \prod_{j\in\mathcal{O}(t)} \Pr\left(\theta_j(t)|\{\zeta_l(t-1)\}_{l\in\Theta_i(t)},\zeta_i(t-1)=x\right)
$$
$$
= \prod_{j\in\mathcal{O}(t)} \Pr\left(\theta_j(t)|\{\zeta_l(t-1)\}_{l\in\partial_j^+(t-1)\setminus\{i\}},\zeta_i(t-1)=x\right).
$$

Based on Assumption 1,

$$
\Pr\left(\{\zeta_l(t-1)\}_{l\in\Theta_i}|\zeta_i(t-1)=x\right) = \prod_{l\in\{\Theta_i(t)\}} \Pr\left(\zeta_l(t-1)\right).
$$

Therefore,

$$
\Pr\left(\underline{Y}(t)|\zeta_i(t-1)=x\right)
$$
$$
= \sum_{\theta_j(t),\ j\in\mathcal{O}(t)} \delta(\{Y_j(t),\theta_j(t)\}_{j\in\mathcal{O}(t)}) \tag{41}
$$
$$
\times \sum_{\zeta_l(t-1)} \prod_{j\in\mathcal{O}(t)} \Pr\left(\theta_j(t)|\{\zeta_l(t-1)\}_{l\in\partial_j^+(t-1)}\setminus\{i\},\zeta_i(t-1)=x\right) \times \prod_{l\in\{\Theta_i(t)\}} \Pr\left(\zeta_l(t-1)\right).
$$

Denote $\{x_j\}_{j\in\mathcal{O}(t)}$ as a realization of $\{\theta_j(t)\}_{j\in\mathcal{O}(t)}$ and $\{y_l\}_{l\in\Theta_i(t)}$ as a realization of $\{\zeta_l(t-1)\}_{l\in\Theta_i(t)}$. Then,

$$
\Pr\left(\underline{Y}(t)|\zeta_i(t-1)=x\right)
$$
$$
= \sum_{\{x_j\}_{j\in\mathcal{O}(t)}} \delta(\{Y_j(t),x_j\}_{j\in\mathcal{O}(t)})
$$
$$
\times \sum_{\{y_l\}_{l\in\Theta_i(t)}} \prod_{j\in\mathcal{O}(t)} \Pr\left(x_j|\{y_l\}_{l\in\partial_j^+(t-1)\setminus\{i\}},\zeta_i(t-1)=x\right) \times \prod_{l\in\{\Theta_i(t)\}} \Pr\left(\zeta_l(t-1)=y_l\right).
$$

Denote

$$
\rho(\{x_j\}_{j\in\mathcal{O}(t)},x)
$$
$$
= \sum_{\{y_l\}_{l\in\Theta_i(t)}} \prod_{j\in\mathcal{O}(t)} \Pr\left(x_j|\{y_l\}_{l\in\partial_j^+(t-1)\setminus\{i\}},\zeta_i(t-1)=x\right) \times \prod_{l\in\Theta_i(t)} \Pr\left(\zeta_l(t-1)=y_l\right). \tag{42}
$$

Then,

$$
\Pr\left(\underline{Y}(t)|\zeta_i(t-1)=x\right) = \sum_{x_j\in\mathcal{X},j\in\mathcal{O}(t)} \delta(\{Y_j(t),x_j\}_{j\in\mathcal{O}(t)})\rho(\{x_j\}_{j\in\mathcal{O}(t)},x). \tag{43}
$$

Based on Assumption 1, we can further simplify (43). Consider node $i$, $\underline{Y}(t)$ can be split into $\underline{Y}_{i,1}(t)$ and $\underline{Y}_{i,2}(t)$, where $\underline{Y}_{i,1}(t)$ is the observations of the set $\mathcal{O}(t)\cap\partial_i^+(t-1)$, and $\underline{Y}_{i,2}(t)$ is the observations of the rest of the nodes. Note that $\underline{Y}_{i,1}(t)\cup\underline{Y}_{i,2}(t)=\underline{Y}(t)$ and $\underline{Y}_{i,1}(t)\cap\underline{Y}_{i,2}(t)=\varnothing$.

**Lemma 3.** *Conditioned on $\underline{Y}_{i,1}(t)$, $\zeta_i(t-1)$ is independent of $\underline{Y}_{i,2}(t)$.*

*Proof.* To show Lemma 3, we use the structured belief network as defined in [70]. $\zeta_j(t)$ is the random variable associated with node $j$. Note that $Y_j(t)$ is the test result of $\zeta_j(t)$ on day $t$. Now, we consider $j\in\left(\mathcal{O}(t)\setminus(\mathcal{O}(t)\cap\partial_i^+(t-1))\right)$. By [70, Theorems 1] and Bayes ball algorithm defined in [71, Section 2], we investigate the following two cases.

(i) For any $j \in \left(\mathcal{O}(t)\backslash(\mathcal{O}(t) \cap \partial_i^+(t-1))\right)$ with $Y_j(t) = 1$, the corresponding state $\zeta_j(t)$ is determined (which is $I$). Then, probabilities conditioning on $Y_j(t)$ is equivalent to (equal to) probabilities conditioning on $\zeta_j(t)$. By Bayes ball algorithm [70, 71], the information (the ball) is blocked at $\zeta_j(t)$ when the information (the ball) reaches $\zeta_j(t)$, which implies the information (the ball) can not reach $\zeta_i(t-1)$.

(ii) For any $j \in \left(\mathcal{O}(t)\backslash(\mathcal{O}(t) \cap \partial_i^+(t-1))\right)$ with $Y_j(t) = 0$, $\zeta_j(t)$ is not determined. By Bayes ball algorithm [70, 71], when the information (the ball) reaches $\zeta_j(t)$, it can traverse $Y_j(t)$ when blocking $Y_j(t)$ (conditioning on $Y_j(t)$). However, by Assumption 1, $\zeta_i(t-1)$ and $\zeta_j(t-1)$ are independent, so any path between $\zeta_i(t-1)$ and $\zeta_j(t-1)$ is blocked, including the path $\zeta_j(t-1) \leftrightarrow \zeta_j(t) \leftrightarrow Y_j(t) \leftrightarrow \zeta_j(t) \leftrightarrow \zeta_i(t-1)$. Thus, the information (the ball) can not reach $\zeta_i(t-1)$.

A simple example is given in Figure 12: Let $Y_1(t) = 0$ and $Y_2(t) = 1$. Given $Y_1(t)$ and $Y_2(t)$, $Y_3(t)$ is independent of $\zeta_1(t-1)$. $\qquad\square$

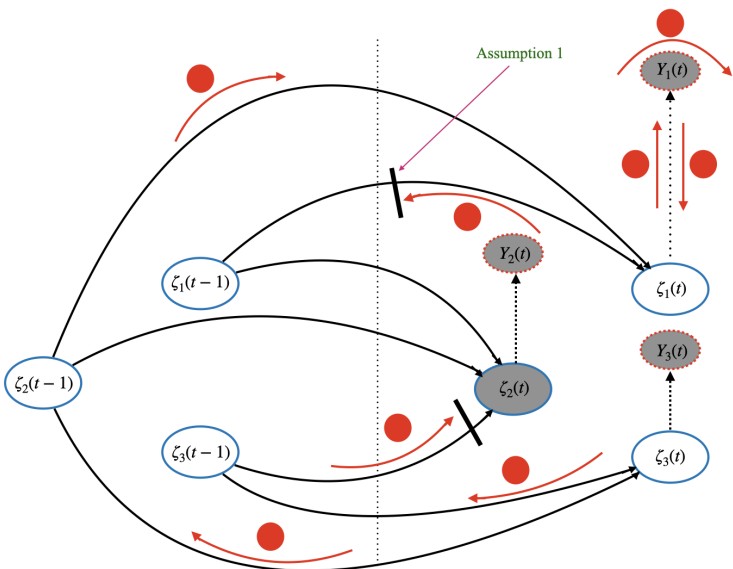

Figure 12: Bayes ball algorithm in the network of 3 nodes. The terms on which we have conditioning are shaded gray and are equivalently blocked.

From Lemma 3,

$$e_x^{(i)}(t-1) = \Pr\left(\zeta_i(t-1) = x | \underline{Y}(t)\right) = \Pr\left(\zeta_i(t-1) = x | \underline{Y}_{i,1}(t)\right). \tag{44}$$

We simplify (43) based on Lemma 3 or (44). From (44), denote the observations of nodes in $\partial_i^+(t-1)$ as $\underline{Y}_{\partial_i^+}(t)$, $\underline{Y}_{\partial_i^+}(t)$ is independent of $\zeta_i(t-1)$. Denote $\Psi_i(t) = \mathcal{O}(t) \cap \partial_i^+(t-1)$. Then, We can replace $\mathcal{O}(t)$ by $\Psi_i(t)$ in (16). Subsequently, denote $\Phi_i(t) = \{j | j \in \partial_k^+(t-1), k \in \Psi_i(t)\}\backslash\{i\}$, and we can replace $\Theta_i(t)$ by $\Phi_i(t)$ in (40). Thus, from (42) and (43), we respectively have

$$\rho\left(\{x_j\}_{j\in\Psi_i(t)}, x\right) = \sum_{\{y_l\}_{l\in\Phi_i(t)}} \prod_{j\in\Psi_i(t)} \Pr\left(x_j | \{y_l\}_{l\in\partial_j^+(t-1)\backslash\{i\}}, \zeta_i(t-1) = x\right) \times \prod_{l\in\Phi_i(t)} \Pr\left(\zeta_l(t-1) = y_l\right) \tag{45}$$

and

$$\Pr\left(\{Y_j(t)\}_{j\in\Psi_i(t)} | \zeta_i(t-1) = x\right) = \sum_{x_j\in\mathcal{X}, j\in\Psi_i(t)} \delta(\{Y_j(t), x_j\}_{j\in\Psi_i(t)}) \rho\left(\{x_j\}_{j\in\Psi_i(t)}, x\right) \tag{46}$$

which give the desired result (17).

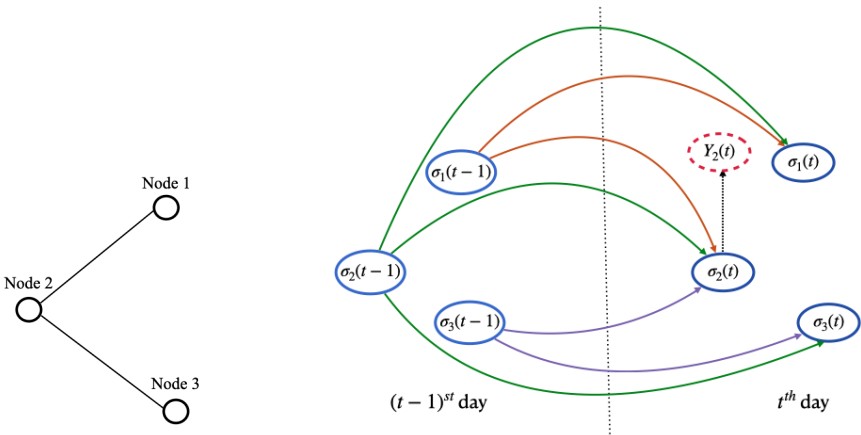

Figure 13: The original graph (left). The graphical model of states and observations (right)

## H  A Simple Example for Algorithm 4

In this section, we give a simple example to illustrate the ideas and steps of Algorithm 4. Besides, we compare our proposed algorithm (Algorithm 4) with the Naive approach discussed in Remark 2. Consider a simple network with three nodes. Node 1 has an edge with node 2, node 2 has an edge with node 3 (see Fig 13). Nodes 1 and 3 are symmetric and statistically identical, and node 2 has higher degree.

Consider the following situation: on the initial day (day 0), assume that nodes 1 and 3 are susceptible, and node 2 is infectious. On day 1, let node 2 be tested. Recall that we define the posterior probability vectors at the end of every day, and the prior probability vectors at the beginning of every day. Nodes' states change in the beginning of every day and testing is also done in the beginning of every day. Let the initial belief, i.e., the posterior probability $\underline{w}_i(0)$ and the prior probability $\underline{u}_i(0)$ on day 0, of the nodes be

$$\underline{w}_i(0) = [1/3, 0, 0, 2/3], \quad i = 1, 2, 3$$
$$\underline{u}_i(0) = [1/3, 0, 0, 2/3], \quad i = 1, 2, 3.$$

**Day** 0: No tests on day 0, the prior probabilities are updated by the forward step (**Step 2** in Algorithm 4).

**Day** 1: By **Step 2** in Algorithm 4, we have

$$\underline{u}_1(1) = [0.3144, 0.0373, 0.0633, 0.5849]$$
$$\underline{u}_2(1) = [0.3559, 0.0676, 0.0633, 0.5131]$$
$$\underline{u}_3(1) = [0.3144, 0.0373, 0.0633, 0.5849].$$

After testing node 2, we know that node 2 is positive. We use the test result to infer about the state of the nodes in prior times. In particular, we update the posterior probability on day 0 ($\underline{w}_i(0)$). Denoting the updated posterior probability as $\underline{e}_i(0)$, by Step 1 in Algorithm 4, we find

$$\underline{e}_1(0) = [0.3144, 0.0373, 0.0633, 0.5849]$$
$$\underline{e}_2(0) = [0.9615, 0.0385, 0.0, 0.0]$$
$$\underline{e}_3(0) = [0.3144, 0.0373, 0.0633, 0.5849].$$

We can now say that at the end of day 0, node 2 was infectious with probability 0.9615 and it was in the latent state with probability 0.0385. Moreover, we see that the (posterior) infection probabilities of nodes 1 and 3 on day 0 have increased since they may have infected node 2 on day 0, i.e., $0.3517 = e_I^{(i)}(0) + e_L^{(i)}(0) > 1/3 = w_I^{(i)}(0) + w_L^{(i)}(0)$ with $i = 1, 3$. Next, we obtain the posterior probability on day 1. Recall that $\underline{w}_i(t)$

describes the posterior probability vector of node $i$ at the end of day $t$. By Step 1 in Algorithm 4,

$$\underline{w}_1(1) = [0.4008, 0.0, 0.1268, 0.4724]$$
$$\underline{w}_2(1) = [0.90, 0.0, 0.10, 0.0]$$
$$\underline{w}_3(1) = [0.4008, 0.0, 0.1268, 0.4724].$$

One may wonder why the posterior probability is $[0.9, 0, 0.1, 0]$ rather than $[1, 0, 0, 0]$. This is because testing is done in the beginning of time $t$ and the posterior probabilities are defined at the end of time slots $t$. The infected node may have recovered by the end of time $t = 1$ and this is reflected in the posterior probabilities computed.

**Day** 2: We can get the prior probability vectors on day 2 by our forward update (making predictions):

$$\underline{u}_1(2) = [0.2883, 0.0, 0.1268, 0.5849]$$
$$\underline{u}_2(2) = [0.8135, 0.0, 0.1865, 0.0]$$
$$\underline{u}_3(2) = [0.2883, 0.0, 0.1268, 0.5849].$$

On the other hand, if we apply the naive updating rule defined in Remark 2, on day 2, we find

$$\underline{u}_1'(2) = [0.4074, 0.0896, 0.0948, 0.4082]$$
$$\underline{u}_2'(2) = [0.81, 0.0, 0.19, 0.0]$$
$$\underline{u}_3'(2) = [0.4074, 0.0896, 0.0948, 0.4082].$$

Recall that we use Assumption 1 in the proposed algorithm (Algorithm 4), and the Naive approach in Remark 2 does not have the backward step, so both approaches do not capture the correlations among nodes. By Monte Carlo simulations, the correlations among nodes are captured, and the nodes' probability vectors are approximated on day 2 as follows:

$$\underline{v}_1(2) = [0.3235, 0.0976, 0.0196, 0.5593]$$
$$\underline{v}_2(2) = [0.7244, 0, 0.2756, 0]$$
$$\underline{v}_3(2) = [0.3158, 0.1072, 0.019, 0.558]$$

which yields the following comparison for the incurred estimation errors:

$$0.4342 = \sum_{i=1}^{3} ||\underline{u}_i(2) - \underline{v}_i(2)|| < \sum_{i=1}^{3} ||\underline{u}_i'(2) - \underline{v}_i(2)|| = 0.5018.$$

The left hand side shows the estimation error under our proposed backward-forward update and the right hand side shows the estimation error under the naive approach.

# I  Delay of Testing Results

One can extend the framework to a more realistic case where testing results are not able to be obtained on the same day, but will be obtained after a delay $a$. In other words, if nodes are tested on day $t - a$, the test results are provided on day $t$. The extended framework is summarized as follows.

On day $t$, before getting the test results of day $t - a$, the algorithm knows the following information: (i) the network topology from day $t - a - 1$ to day $t$, i.e., $\mathcal{G}(t - a - 1), \cdots, \mathcal{G}(t)$ (it is affected by the past actions); (ii) the posterior probability of nodes on day $t - a - 1$, $\{\underline{w}_i(t - a - 1)\}_{i \in \mathcal{G}(t-a-1)}$; and (iii) and the prior probability vectors of nodes from day $t - a$ to day $t$, i.e., $\{\underline{u}_i(t - a)\}_{i \in \mathcal{G}(t-a)}, \cdots, \{\underline{u}_i(t)\}_{i \in \mathcal{G}(t)}$.

After getting the test results on day $t - a$, we can obtain the updated posterior probability vectors on day $t - a - 1$, and the posterior probability vectors on day $t - a$, i.e., $\{\underline{e}_i(t - a - 1)\}_{i \in \mathcal{G}(t-a-1)}$, and $\{\underline{w}_i(t - a)\}_{i \in \mathcal{G}(t-a)}$, by **Step 1** in Algorithm 4.

Based on $\{\underline{w}(t-a)\}_{i\in\mathcal{G}(t-a)}$, by **Step 2** in Algorithm 4, we update the prior probability from day $t-a+1$ to day $t$, and obtain the prior probability on day $t+1$, i.e., $\{\underline{u}_i(t+1)\}_{i\in\mathcal{G}(t+1)}$.

Repeating the process, we can compute the estimated probability vectors of nodes and apply the exploration and exploitation policies.

## J   Proof of Theorem 2

**Step 1**: Preliminaries.

We divide the distributions of initial infectious nodes into two complementary events:

$$\mathcal{I}_1 = \{\text{No node is infectious}\}$$
$$\mathcal{I}_2 = \mathcal{I}_1^c.$$

Let $N$ be sufficiently large,

$$\Pr\{\mathcal{I}_1\} = (1-1/N)^N \approx 1/e$$
$$\Pr\{\mathcal{I}_2\} \approx 1 - 1/e.$$

In event $\mathcal{I}_1$, since there is no infection on the initial day, then no node is infectious in the future, i.e., the true probability of nodes $v_I^{(i)}(t) = 0$ for all $i \in \mathcal{V}(t)$ and $t \geq 1$.

Note that in Example 1, each node can be in one of two states, $S$ and $I$. The transmission probability $\beta = 1$. So, on day $t$, the probability of node $i$ being in state $I$ includes the infection of node $i$ on day $t-1$, and the infection from its neighbors. Then, based on (13), we have

$$
\begin{aligned}
u_I^{(i)}(t) &= w_I^{(i)}(t-1) + \{1 - w_I^{(i)}(t-1)\}\Big\{1 - \big(1 - w_I^{(i-1)}(t-1)\big)\big(1 - w_I^{(i+1)}(t-1)\big)\Big\} \\
&= 1 - \{1 - w_I^{(i-1)}(t-1)\}\{1 - w_I^{(i)}(t-1)\}\{1 - w_I^{(i+1)}(t-1)\}.
\end{aligned}
\tag{47}
$$

For convention, we assume that nodes $0$ and $N+1$ are two virtual nodes with no probability of infection, i.e., $u_I^{(0)}(t) = u_I^{(N+1)}(t) = 0$ for all $t$, and no tests are applied to these two nodes all the time.

Since $w_I^{(i)}(t+1), w_I^{(i-1)}(t+1), w_I^{(i+1)}(t+1) \in [0,1]$, then from (47),

$$u_I^{(i)}(t) \geq 1 - 1 \times (1 - w_I^{(i)}(t-1)) \times 1 = w_I^{(i)}(t-1). \tag{48}$$

Thus, by symmetry over $w_I^{(i-1)}(t-1)$, $w_I^{(i)}(t-1)$, and $w_I^{(i+1)}(t-1)$ we get the inequality

$$u_I^{(i)}(t) \geq \max\{w_I^{(i-1)}(t-1), w_I^{(i)}(t-1), w_I^{(i+1)}(t-1)\}. \tag{49}$$

**Step 2**: Consider the computation of $\{\underline{u}_i(t)\}_i$ based on (13) $\big($equivalently (47)$\big)$ under event $\mathcal{I}_1$.

Recall that $B(t) = 1$ for all $t$. On any day $t$, if node $i_0$ is tested, then the result is negative, and $w_I^{(i_0)}(t) = 0$, and

$$w_I^{(i)}(t) = u_I^{(i)}(t) \text{ for all } i \neq i_0. \tag{50}$$

In (49), at most one of $w_I^{(i-1)}(t-1)$, $w_I^{(i)}(t-1)$, and $w_I^{(i+1)}(t-1)$ is updated to $0$. We first prove the following facts.

**Fact 1**. $u_I^{(i)}(t) \geq \frac{1}{N}$ for all $t$. On any day $t$, $w_I^{(i)}(t) \geq \frac{1}{N}$ with $i \neq i_0$, where $i_0$ is the index of node tested on day $t$.

*Proof.* We prove **Fact 1** by mathematical induction. On the initial day, by model assumption in Example 1, $u_I^{(i)}(0) = \frac{1}{N}$ for all $i$. Then, if node $i_0$ is tested, then as mentioned above, $w_I^{(i_0)}(0) = 0$, and by (50) $w_I^{(i)}(0) = u_I^{(i)}(0) = 1/N$ for all $i \neq i_0$.

Suppose **Fact 1** holds for all $\tau \leq t-1$. Now, we consider $\tau = t$. From (49), we have $u_I^{(i)}(t) \geq \max\{w_I^{(i-1)}(t-1), w_I^{(i)}(t-1), w_I^{(i+1)}(t-1)\} \geq 1/N$. Then, if node $i_0$ is tested, we have $w_I^{(i_0)}(t) = 0$, and then by (50), $w_I^{(i)}(t) = u_I^{(i)}(t) \geq 1/N$ for all $i \neq i_0$. $\qquad \square$

**Fact 2**. If node $i$ has not been tested up to day $t$, then $u_I^{(i)}(t)$ tends to 1 as $t \to \infty$.

*Proof.* Since node $i$ is not tested from the initial day to day $t$, then

$$w_I^{(i)}(\tau) = u_I^{(i)}(\tau), \; \tau \leq t. \tag{51}$$

Note that at most one of its neighbors is tested on day $t$. By (47) and **Fact 1**,

$$u_I^{(i)}(t) \geq 1 - (1 - 1/N)(1 - w_I^{(i)}(t-1)) = 1 - (1 - 1/N)(1 - u_I^{(i)}(t-1)),,$$

which implies

$$(1 - 1/N)(1 - u_I^{(i)}(t-1)) \geq 1 - u_I^{(i)}(t),$$

which implies

$$1 - u_I^{(i)}(t) \leq (1 - 1/N)^t(1 - u_I^{(i)}(0)) = (1 - 1/N)^{t+1}.$$

Letting $t \to \infty$ completes the proof. $\qquad \square$

**Fact 3**. If node $i$ is not tested on day $t-1$, then

$$u_I^{(i)}(t) \geq w_I^{(i)}(t-1) + \frac{1}{N}(1 - w_I^{(i)}(t-1))w_I^{(i)}(t-1).$$

*Proof.* By **Fact 3**, if node $i$ is not tested on day $t-1$, then $w_I^{(i)}(t-1) > 0$. From (47), by some algebra,

$$\begin{aligned} u_I^{(i)}(t) =& w_I^{(i)}(t-1) + (1 - w_I^{(i)}(t-1))(w_I^{(i-1)}(t-1) + w_I^{(i+1)}(t-1) - w_I^{(i-1)}(t-1)w_I^{(i+1)}(t-1)) \\ =& (1 + \epsilon)w_I^{(i)}(t-1) \end{aligned}$$

where

$$\epsilon = \frac{1 - w_I^{(i)}(t-1)}{w_I^{(i)}(t-1)} \times (w_I^{(i-1)}(t-1) + w_I^{(i+1)}(t-1) - w_I^{(i-1)}(t-1)w_I^{(i+1)}(t-1)).$$

Note that at most one of the neighbors of node $i$ is tested on day $t-1$, then

$$\frac{1 - w_I^{(i)}(t-1)}{w_I^{(i)}(t-1)} \geq 1 - w_I^{(i)}(t-1)$$

$$w_I^{(i-1)}(t-1) + w_I^{(i+1)}(t-1) - w_I^{(i-1)}(t-1)w_I^{(i+1)}(t-1) \geq \max\{w_I^{(i-1)}(t-1), w_I^{(i+1)}(t-1)\}.$$

From **Fact 1**, $\max\{w_I^{(i-1)}(t-1), w_I^{(i+1)}(t-1)\} \geq 1/N$. Thus, $\epsilon \geq (1 - w_I^{(i)}(t-1)) \times 1/N$. Hence, $u_I^{(i)}(t) \geq w_I^{(i)}(t-1) + \frac{1}{N}(1 - w_I^{(i)}(t-1))w_I^{(i)}(t-1)$. $\qquad \square$

Since we consider all possible sequential testing policies, then we divide all nodes into two sets

$$\begin{aligned} \mathcal{S}_1(t) =& \{\text{nodes that have not been tested up to day } t\} \\ \mathcal{S}_2(t) =& \mathcal{S}_1^c(t). \end{aligned}$$

In the following proof, let $t \to \infty$. By **Fact 2**, $u_I^{(i)}(t) \to 1$ if $i \in \mathcal{S}_1(t)$. Next, we focus on the set $\mathcal{S}_2(t)$. Denote the index of node which is tested on day $t-1$ as $i_0(t)$. By **Fact 1**, $w_I^{(i)}(t-1) \geq 1/N$ for all $i \neq i_0(t)$. Then, we define

$$\begin{aligned} \mathcal{S}_{21}(t) =& \{i | 1/N \leq w_I^{(i)}(t-1) < 1 - 1/N\} \\ \mathcal{S}_{22}(t) =& \{i | 1 - 1/N \leq w_I^{(i)}(t-1)\}. \end{aligned}$$

Thus, we have $\mathcal{S}_2(t) = \mathcal{S}_{21}(t) \cup \mathcal{S}_{22}(t) \cup \{i_0(t)\}$. Due to the equivalence of norms, without loss of generality, we consider $L_1$ norm in the rest of the proof.

(i) If $i \in \mathcal{S}_1(t)$, then $u_I^{(i)}(t) \to 1$. Thus $||\underline{u}_i(t) - \underline{v}_i(t)||_1 \to 2$.

(ii) If $i \in \mathcal{S}_{21}(t)$, then $||\underline{u}_i(t) - \underline{v}_i(t)||_1 \geq ||\underline{u}_i(t-1) - \underline{v}_i(t-1)||_1 + \frac{2(N-1)}{N^3}$. In fact, since $i \in \mathcal{S}_{21}(t)$, then $i \neq i_0(t)$, thus by (50) and **Fact 3**,

$$u_I^{(i)}(t) \geq u_I^{(i)}(t-1) + \frac{1}{N}(1 - w_I^{(i)}(t-1))w_I^{(i)}(t-1).$$

Note that $N$ is sufficiently large, so $1/N < 1/2 < 1 - 1/N$. If $x \in [1/N, 1-1/N)$, then the fuction $f(x) = x(1-x)$ has the minimum value $\frac{N-1}{N^2}$ when $x = 1/N$. Thus,

$$u_I^{(i)}(t) \geq u_I^{(i)}(t-1) + \frac{N-1}{N^3}. \tag{52}$$

Recall that $v_I^{(i)}(t) = 0$ and $v_S^{(i)}(t) = 1$ for all $t$, and $u_I^{(i)}(t) + u_S^{(i)}(t) = 1$, then

$$||\underline{u}_i(t) - \underline{v}_i(t)||_1 = |u_I^{(i)}(t) - v_I^{(i)}(t)| + |u_S^{(i)}(t) - v_S^{(i)}(t)| = 2|u_I^{(i)}(t) - v_I^{(i)}(t)|. \tag{53}$$

From (52),

$$||\underline{u}_i(t) - \underline{v}_i(t)||_1 = 2|u_I^{(i)}(t) - v_I^{(i)}(t)| \geq 2|u_I^{(i)}(t-1) + \frac{N-1}{N^3} - v_I^{(i)}(t-1)|$$
$$\geq 2|u_I^{(i)}(t-1) - v_I^{(i)}(t-1)| + \frac{2(N-1)}{N^3} = ||\underline{u}_i(t-1) - \underline{v}_i(t-1)||_1 + \frac{2(N-1)}{N^3}.$$

(iii) If $i \in \mathcal{S}_{22}(t)$, then node $i$ is not tested on day $t$, thus from (49), $u_I^{(i)}(t) \geq w_I^{(i)}(t-1) = 1 - 1/N$. Thus, by (53), $||\underline{u}_i(t) - \underline{v}_i(t)||_1 \geq \frac{2(N-1)}{N}$.

Since we consider $N$ sufficiently large, then we can prove the following lemma.

**Lemma 4.** $\lim_{t \to \infty} \mathcal{S}_{21}(t) = \varnothing$.

*Proof.* We first prove the following Claims.

**Claim 1**. If (i) $u_I^{(i-1)}(t) \geq 1 - 1/N$ and node $i-1$ is not tested on day $t$, or (ii) $u_I^{(i+1)}(t) \geq 1 - 1/N$ and node $i+1$ is not tested on day $t$, or (iii) $u_I^{(i-1)}(t) \geq 1 - 1/N$ and $u_I^{(i+1)}(t) \geq 1 - 1/N$, then $u_I^{(i)}(t+1) \geq 1 - 1/N$.

**Proof**. By (47) and (50), we can derive $u_I^{(i)}(t+1) \geq 1 - 1/N$ directly. $\square$

**Claim 2**. No node can stay in $\mathcal{S}_{21}(t)$ for successive $\lceil N^3/(N-1) \rceil$ days.

**Proof**. if node $i$ stays in $\mathcal{S}_{21}(t)$ for successive $\lceil N^3/(N-1) \rceil$ days, i.e., from day $\tau$ to day $\tau + \lceil N^3/(N-1) \rceil$, then by (52), $u_I^{(i)}(\tau + \lceil N^3/(N-1) \rceil) > 1$, which contradicts with $u_I^{(i)}(t) \leq 1$ for all $t$. $\square$

Now, we prove the lemma by contradiction. Based on **Claim 2**, assume there exists at least one $j$ and an increasing sequence $\{t_i\}_{i=0}^{\infty}$ with $\lim_{n \to \infty} t_n = \infty$, such that $j \in \mathcal{S}_{21}(t_i)$ for all $\{t_i\}_{i=0}^{\infty}$.

For some $i$, node $j$ is in $\mathcal{S}_{22}(t_i - 1)$ on day $t_i - 1$, and node $j$ is in $\mathcal{S}_{21}(t_i)$ on day $t_i$. In other words, $u_I^{(j)}(t_i) < 1 - 1/N \leq u_I^{(j)}(t_i - 1)$. From (47) and **Calim 1**, $u_I^{(j)}(t_i) < 1 - 1/N \leq u_I^{(j)}(t_i - 1)$ holds only because node $j$ is tested on day $t_i - 1$, and all of its neighbors (i.e., nodes $j-1$, $j+1$) have $u_I^{(j-1)}(t_i - 1) < 1 - 1/N$ and $u_I^{(j+1)}(t_i - 1) < 1 - 1/N$. However, since $u_I^{(j)}(t_i - 1) \geq 1 - 1/N$ and node $j$ is tested on day $t_i - 1$, then by **Claim 1**, $u_I^{(j-1)}(t_i) \geq 1 - 1/N$ and $u_I^{(j+1)}(t_i) \geq 1 - 1/N$. Subsequently, by **Claim 1**, we have $u_I^{(j)}(t_i+1) \geq 1 - 1/N$. Thus, on day $t_i + 1$, at least one of its neighbors, say $j-1$, has $u_I^{(j-1)}(t_i+1) \geq 1 - 1/N$. By **Claim 1**, node $j$ never fall into $\mathcal{S}_{21}(t)$ for $t \in \{t_{i+1}, t_{i+2}, \cdots\}$, which contradicts with the assumption. $\square$

From Lemma 4, when $t \to \infty$, we have $|\mathcal{S}_1(t)| = \Theta(N)$ or $|\mathcal{S}_{22}(t)| = \Theta(N)$. Thus, $\sum_{i=1}^{N} ||\underline{u}_i(t) - \underline{v}_i(t)||_1 = \Theta(N)$.

**Step 3**: Consider the computation of $\{\underline{u}_i(t)\}_i$ based on Algorithm 4.

In this step, we consider a specific testing policy: We test node $i$ on day $k$, where $k \equiv i - 1 (mod\ M)$ for all $1 \le i \le M$.

In event $\mathcal{I}_2$, since the transmission probability $\beta = 1$, then all nodes are infected at most $N$ days because there is no recovery. Thus, no node with positive testing result is repeatedly tested. So in at most $2N$ days, all nodes are infectious, and the algorithm finds all infected nodes, so $\underline{u}_i(t) = \underline{v}_i(t)$, $t \ge 2N$.

In event $\mathcal{I}_1$, whenever a node is tested, it is negative. Node 1 is tested on day 0, the result is negative. On day 1, node 2 is tested and the result is negative. By backward updating, since $\beta = 1$ and no recovery, then nodes 1&3 are inferred to be in state $S$ on day 0. Since node 2 is in state $S$ on day 1. Then, node 1 is inferred in state $S$ on days 0 and 1.

Assume that nodes $1, 2, \cdots, k - 2$ are inferred to be in state $S$ by day $k - 1$. Now, we day $k$, where $k \le N$. On day $k$, node $k - 1$ is tested negative, hence by backward updating, nodes $k - 2$ and $k$ are inferred to be in state $S$ on day $k - 1$. By the testing result of node $k - 1$ on day $k$, nodes $1, 2, \cdots, k - 1$ are inferred in state $S$ by day $k$. By induction, after $N$ days, it clears every node, so $\underline{u}_i(t) = \underline{v}_i(t)$, $t \ge N$.

From **Steps 1∼3**, we complete the proof.

## K  $\alpha$-linking Backward Updating

### K.1  Complexity Reduction

Let $\{x_j\}_{j \in \mathcal{O}(t)}$ be a realization of $\{\theta_j(t)\}_{j \in \mathcal{O}(t)}$ and $\{y_l\}_{l \in \Theta_i(t)}$ be a realization of $\{\zeta_l(t - 1)\}_{l \in \Theta_i(t)}$. Let node $i$ have state $x$ in day $t - 1$. Consider one node $k \in \partial_j^+(t - 1)\backslash\{i\}$ and the probability

$$\Pr\left(x_j | \{y_l\}_{l \in \partial_j^+(t-1)\backslash\{i\}}, x\right), j \in \Psi_i(t).$$

Since node $k$ is not infectious if $y_k = L$, $y_k = R$ or $y_k = S$, then the probability above remians the same no matter whether $y_k = L$, $y_k = R$ or $y_k = S$.

Thus, we introduce a new state, denoted by $E$, to be a replacement of $\{L, R, S\}$, and

$$\Pr\left(y_k = E\right) = \sum_{x \in \{L,R,S\}} \Pr\left(y_k = x\right).$$

Next, denote $\mathcal{X}' = \{I, E\}$. Equation (17) can be re-written as follows:

$$
\begin{aligned}
&\Pr\left(\{Y_j(t)\}_{j \in \Psi_i(t)} | \zeta_i(t - 1) = x\right) \\
&= \sum_{\{x_j\}_{j \in \Psi_i(t)}} \prod_{j \in \Psi_i(t)} \Pr\left(Y_j(t) | \theta_j(t)\right) \times \sum_{\{y_l\}_{l \in \Theta_i(t)}} \prod_{j \in \Psi_i(t)} \mathsf{P}_j\left(x_j | \{y_l\}_{l \in \partial_j^+(t-1)\backslash\{i\}}, x\right) \\
&\quad \times \prod_{z_l \in \mathcal{X}', l \in \Theta_i(t)} \Pr\left(\zeta_l(t - 1) = z_l\right),
\end{aligned}
\tag{54}
$$

with reduces the computation complexity. Subsequently, $\underline{e}_i(t - 1)$ in (16) can be calculated by (54) directly.

### K.2  $\alpha$-linking Backward Updating

In the backward step, the computation complexity is large even in (54). To further reduce the complexity in (54), one way is to update the posterior probability $\underline{e}_i(t)$ in a sparser network. Now, we define $\alpha$-linking Backward Updating as follows:

(i) We generate a subgraph $\mathcal{G}_\alpha(t)$ based on the pre-determined graph $\mathcal{G}(t)$: Suppose that each edge (in $\mathcal{G}(t)$) exists with probability $\alpha$, $0 \le \alpha \le 1$. If $\alpha = 1$, then $\mathcal{G}_\alpha(t) = \mathcal{G}(t)$; if $\alpha = 0$, then $\mathcal{G}_\alpha(t)$ is a graph with no edges.

(ii) Backward updating in $\mathcal{G}_\alpha(t)$: Similar with $\partial_i(t)$, $\Psi_i(t)$, $\Phi_i(t)$ and $\Theta_i(t)$, we define $\partial_{i,\alpha}(t)$, $\Psi_{i,\alpha}(t)$, $\Phi_{i,\alpha}(t)$ and $\Theta_{i,\alpha}(t)$ on graph $\mathcal{G}_\alpha(t)$, respectively. Subsequently, replace $\partial_i(t)$, $\Psi_i$, $\Phi_i(t)$ and $\Theta_i(t)$ by $\partial_{i,\alpha}(t)$, $\Psi_{i,\alpha}(t)$, $\Phi_{i,\alpha}(t)$ and $\Theta_{i,\alpha}(t)$ in (54), respectively.

## L  Proof of Theorem 3

**Step 1**. Preliminaries.

In Example 2, $\beta = 1$, $\lambda = 0$, and $\gamma = 0$, there is no recovery and we assume no latent state. Based on (9), the expression of rewards $\hat{r}_i(t)$ for every node is given as follows. If node $i$ has two neighbors (without quarantine)

$$\hat{r}_i(t) = u_S^{(i-1)}(t)(1 - u_I^{(i-2)}(t))u_I^{(i)}(t) + u_S^{(i+1)}(t)(1 - u_I^{(i+2)}(t))u_I^{(i)}(t). \tag{55}$$

If node $i$ only has one neighbor, then

$$\hat{r}_i(t) = u_S^{(i+d)}(t)\big(1 - u_I^{(i+2d)}(t)\big)u_I^{(i)}(t), \ d \in \{-1, 1\}. \tag{56}$$

For simplicity, we introduce artificial nodes $-1, 0, N+1, N+2$ with $u_I^{(-1)}(t) = u_I^{(0)}(t) = u_I^{(N+1)}(t) = u_I^{(N+2)}(t) = 0$ for all $t$, and these 4 nodes are never tested.

**Step 2**. The RbEx policy.

Under the RbEx policy, the algorithm always tests the nodes with maximum rewards. Let an infectious node be found, for the first time, on day $aN$, where $a$ is a positive real number. Note that until the first infected node is found, in any application of the RbEx policy, $u_I^{(i)}(t)$ is the same for any given $i$, and hence $\hat{r}_i(t)$ is also the same. So, $a$ is the same for any application of the RbEx policy. Recall that in Example 2, nodes that are tested positive will be isolated. The cumulative infections is at least $\min\{aN, N\}$ in the end.

**Step 3**. Consider the exploration process of the specific exploration policy.

Recall that from **Step 2**, an infectious node is found, for the first time, by the RbEx policy with budget 10 tests on day $aN$. Under the specific defined exploration policy, we can choose a specific $b'$ with $b' < a$, such that no infectious node is tested by the RbEx policy with budget 9 tests before and including day $t = b'N$.

We know that on day $\tau$, nodes $1, 2, \cdots, \tau$ are infectious since $\beta = 1$. Note that one test is applied to exploration (randomly choice) on every day, so with probability

$$\prod_{\tau'=1}^{\tau}(1 - \frac{\tau'}{N}), \tag{57}$$

no infectious node is explored from the initial day to day $\tau$. Then, with probability

$$\prod_{\tau'=1}^{\tau-1}(1 - \frac{\tau'}{N}) \cdot \frac{\tau}{N},$$

one infectious node is detected on day $\tau$. Thus, with probability

$$\sum_{\tau=1}^{t}\prod_{\tau'=1}^{\tau-1}(1 - \frac{\tau'}{N}) \cdot \frac{\tau - 1}{N}, \tag{58}$$

one infectious node is tested by exploration process on day $\tau$ ($\tau \leq t$), and this node is not the new infectious one on day $\tau$, i.e., has index $\tau$. The probability defined in (58) increases with $t$ when $N$ is fixed, and it can be close to 1 when $t$ close to $N$. Therefore, We can choose proper parameters $b'$ and $N$ such that the probability defined in (58) is larger than or equal to $p_0$. In particular, if $N$ is large, we can choose a relatively small $b'$. In Theorem 3, we set $p_0 \geq 99/100$.

Let the infectious node detected (for the first time) by the exploration process have index $j$ on day $t'$, where $t' \leq t$. As discussed above, node $j$ is not the new infectious node on day $t'$, so we have $j < t'$. In other words, node $j + 1$ must be infeicotus on day $t'$ with a positive test result, i.e., $Y_j(t') = 1$. By Step 1 in Algorithm 4, the updated posterior probability of node $j$

$$e_I^{(j)}(t' - 1) = 1, \quad e_S^{(j)}(t' - 1) = 0. \tag{59}$$

Again, by Step 1 in Algorithm 4,

$$w_I^{(j-1)}(t') = w_I^{(j+1)}(t') = 1. \tag{60}$$

Then, by Step 2 in Algorithm 4,

$$u_I^{(j-2)}(t'+1) = u_I^{(j-1)}(t'+1) = u_I^{(j)}(t'+1) = u_I^{(j+1)}(t'+1) = u_I^{(j+2)}(t'+1) = 1. \tag{61}$$

Since $j$ is detected and isolated on day $t'$, then,

$$\hat{r}_j(t'+1) = 0. \tag{62}$$

By (56) and (61),

$$\hat{r}_{j-1}(t'+1) = u_S^{(j-2)}(t'+1)\big(1 - u_I^{(j-3)}(t'+1)\big) = 0$$
$$\hat{r}_{j+1}(t'+1) = u_S^{(j+2)}(t'+1)\big(1 - u_I^{(j+3)}(t'+1)\big) = 0. \tag{63}$$

By (55) and (61),

$$\hat{r}_{j-2}(t'+1) = u_S^{(j-3)}(t'+1)\big(1 - u_I^{(j-4)}(t'+1)\big)$$
$$\hat{r}_{j+2}(t'+1) = u_S^{(j+3)}(t'+1)\big(1 - u_I^{(j+4)}(t'+1)\big). \tag{64}$$

**Step 4.** The exploitation process of the specific exploration policy.

We first study an extreme case where no tests are applied. In this case, denote the prior probability of node $i$ on day $\tau$ as $U_I^{(i)}(\tau)$, which can be calculated by the following recursion:

$$U_I^{(i)}(\tau+1) = U_I^{(i)}(\tau) + U_S^{(i)}(\tau)\big(1 - (1 - U_I^{(i-1)}(\tau))(1 - U_I^{(i+1)}(\tau))\big). \tag{65}$$

Based on (65), recall that $U_I^{(i)}(0) = 0$ if $i \leq \frac{9N}{10}$, and $U_I^{(i)}(0) = \frac{10\epsilon}{N}$ if $\frac{9N}{10} < i \leq N$, then $U_I^{(i)}(\tau)$ increases over $\tau$ and is a function of $\epsilon$. Then, given $b'$, $N$ and $t = b'N$, we can choose a small enough $\epsilon$, denoted by $\epsilon(b', N)$, such that $U_I^{(i)}(2t) < \frac{1}{2}$ for all $i$. Since $U_I^{(i)}(\tau)$ increases over $\tau$, then $U_I^{(i)}(\tau) < \frac{1}{2}$, $\tau \leq 2t$.

Now, we introduce the exploitation process. Let $t = b'N < \min\{\frac{9}{40}, a\}N$. There are at most $2t$ infectious nodes on day $2t$, i.e., nodes $1, 2, \cdots, 2t$. Since $t < \min\{\frac{9}{40}, a\}N$, then nodes with index from $9N/10 - 2t$ to $N$ are in state $S$, which implies nodes with index from $9N/10 - 2t$ to $N$ can never be tested positive before day $2t$. Thus, on any day $\tau \leq 2t$, for $9N/10 - 2t \leq i \leq N$, if node $i$ is tested, and the testing result is negative. Recall that $U_I^{(i)}(\tau)$ in (65) is calculated without any negative testing results. Hence, $u_I^{(i)}(\tau) \leq U_I^{(i)}(\tau)$. Furthermore, with the condition $t = b'N < \min\{\frac{9}{40}, a\}N$, we can find a small enough $\epsilon(b', N)$, such that under the specific exploration policy,

$$u_I^{(i)}(\tau) < \frac{1}{2}, \quad \tau \leq 2t, \ 9N/10 - 2t \leq i \leq N. \tag{66}$$

In the rest, we divide the nodes in to 3 sets: $\mathcal{Q}_1 = \{i | i \leq 2t\}$, $\mathcal{Q}_2 = \{i | 2t < i < 9N/10 - 2t\}$, and $\mathcal{Q}_3 = \{i | 9N/10 - 2t \leq i \leq N\}$.

**Fact 1.** For $i \in \mathcal{Q}_1$ and $\tau \leq 2t$, $u_I^{(i)}(\tau) = 1$ or $u_I^{(i)}(\tau) = 0$.

*Proof.* If no test is applied to $\mathcal{Q}_1$, then $u_I^{(i)}(\tau) = 0$ for all $i \in \mathcal{Q}_1$.

On some day $\tau \leq 2t$, if one node with index $j \in \mathcal{Q}_1$ is tested positive on day $\tau - 1$, then by (61), $u_I^{(j-2)}(\tau) = u_I^{(j-1)}(\tau) = u_I^{(j)}(\tau) = u_I^{(j+1)}(\tau) = u_I^{(j+2)}(\tau) = 1$. In other words, if node $j$ is tested positive on day $\tau - 1$, then node $j$, its neighbors and neighbors of neighbors have probability of infection equal to 1 on day $\tau$.

On some day $\tau$, if node $j$ is not tested positive on day $\tau - 1$, and neither of its neighbors and neighbors of neighbors are is not tested positive, then $u_I^{(j)}(\tau) = 1$ only when $u_I^{(j)}(\tau - 1) = 1$, or $u_I^{(j-1)}(\tau - 1) = 1$ or $u_I^{(j+1)}(\tau - 1) = 1$ since $\beta = 1$. Otherwise $u_I^{(j)}(\tau) = 0$. $\square$

**Fact 2**. For $i \in \mathcal{Q}_1$ and $\tau \leq 2t$, $\hat{r}_i(\tau) = 1$ or $\hat{r}_i(\tau) = 0$.

*Proof.* If $u_I^{(i)}(\tau) = 0$, then $\hat{r}_i(\tau) = 0$ by (55) and (56).

Now, we consider $u_I^{(i)}(\tau) = 1$ in the following cases: (i) If both neighbors of node $i$ are isolated, then $\hat{r}_i(\tau) = 0$. (ii) If one of neighbors of node $i$ (for example, node $i - 1$) is isolated, then by (56), $\hat{r}_i(\tau) = 0$ when $u_I^{(i+1)}(\tau) = 1$, and $\hat{r}_i(\tau) = 1$ when $u_I^{(i+1)}(\tau) = 0$. (iii) If both neighbors are not isolated, then $u_I^{(i-1)}(\tau - 1) = 1$ or $u_I^{(i+1)}(\tau - 1) = 1$, otherwise, $u_I^{(i)}(\tau) = 0$. Since there is no recovery, then $u_I^{(i-1)}(\tau) = 1$ or $u_I^{(i+1)}(\tau) = 1$. By **Fact 1**, $u_I^{(j)}(\tau) = 1$ or $u_I^{(j)}(\tau) = 0$ when $j \in \mathcal{Q}_1$. If $u_I^{(i+1)}(\tau) = u_I^{(i-1)}(\tau) = 1$, then $\hat{r}_i(\tau) = 0$. If $u_I^{(i+1)}(\tau) = 1$, then by (55), $\hat{r}_i(\tau) = 0$ when $u_I^{(i-2)}(\tau) = 1$, $\hat{r}_i(\tau) = 1$ when $u_I^{(i-2)}(\tau) = 0$. If $u_I^{(i-1)}(\tau) = 1$, then by (55), $\hat{r}_i(\tau) = 0$ when $u_I^{(i+2)}(\tau) = 1$, $\hat{r}_i(\tau) = 1$ when $u_I^{(i+2)}(\tau) = 0$. □

From (66), for all $\tau \leq 2t$ and $i \in \mathcal{Q}_3$, we have

$$\hat{r}_i(\tau) \leq 2u_I^{(i)}(\tau) < 1. \tag{67}$$

Note that only nodes in $\mathcal{Q}_1$ and $\mathcal{Q}_3$ may have positive probability of infection. For all $\tau \leq 2t$ and $i \in \mathcal{Q}_2$, since $u_I^{(i)}(\tau) = 0$, then $\hat{r}_i(\tau) = 0$. Therefore, a node with reward equal to 1 has the largest reward.

Recall that on day $t'$, node $j$ is tested positive. From (64), nodes $j - 2$ and $j + 2$ have largest rewards ($= 1$) on day $t' + 1$, which are exploited on day $t' + 1$, and all other nodes in $\mathcal{Q}_1$ have rewards 0. This is because $t'$ is the first day when a positive node is found. Since node $j$ is tested positive and isolated on day $t'$, then all infectious nodes with index less than $j$ can no longer infect other nodes in the line network. Now, we consider the nodes with index larger than $j$. Recall that $j < t'$, so node $j + 1$ must be infectious on day $t'$, and node $j + 2$ must be infectious on day $t' + 1$ since $\beta = 1$. Thus, node $j + 2$ is tested positive and is isolated. Since the network is a line, both nodes $j + 1$ and $j + 2$ can no longer infect other nodes once node $j + 2$ is isolated. Note that nodes in $\mathcal{Q}_3$ have positive rewards. When $N$ is sufficiently large, in the rest of the exploitation process, nodes in $\mathcal{Q}_3$ are tested. Recall that we have one test for exploration, and we can isolate at least 2 infectious nodes with index larger than $j$.

Repeat the process, we exploit nodes $j + 4, j + 6, \cdots$ on day $t' + 2, t' + 3, \cdots$, respectively. Consider the direction from node 1 to node $N$. On every day, there is at most one new infectious node, but at least two infectious nodes can be isolated. On some day, denote as day $t' + x$, the exploitation process can progress beyond the infections (exceeding by one node) for the first time. In other words, node $j + 2x$ is tested negative on day $t' + x$. By Step 2 in Algorithm 4, $e_I^{(j+2x)}(t' + x - 1) = 0$. However, since $w_I^{(j+2x-1)}(t' + x) = 1$ becuase node $j + 2x - 2$ is tested positive on day $t' + x - 1$. By Step 1 in Algorithm 4, $u_I^{(j+2x)}(t' + x + 1) = 1$, hence by (64), $\hat{r}_{j+2x}(t' + x + 1) = 1$, which implies node $j + 2x$ has the largest reward and is exploited on day $t' + x + 1$, and it will be tested positive. On day $t' + x + 1$, all infectious nodes are isolated.

Finally, we can calculate the total number of infections to be

$$j + 2(t' - j) = 2t' - j \leq 2t' \leq 2t = 2b'N.$$

Let $b = 2b'$. This is an improvement by a factor of at least $\frac{a}{b}$ in comparison to the RbEx strategy, where $\frac{a}{b}$ can be as large as desired by increasing the value of $N$ or decreasing $p_0$.

## M  Construction of Networks and Further Results

### M.1  Constructions of SBM and V-SBM

In this section, we construct SBMs and its variants.

**SBM**  The SBM is a generative model for random graphs. The graph is divided into several communities, and subsets of nodes are characterized by being connected with one another with particular edge densities.[8] The intra-connection probability is $p_1$, and inter-connection probability is $p_2$. We denote the SBM as

---

[8] Here, we assume that $M$ is an exact divisor of $N$.

| WS, $(d, \delta)$ | $\gamma_c$ | $L_p$ | Ratio |
|---|---|---|---|
| $(6, 0.05)$ | **0.504** | 4.952 | .0003 |
| $(4, 0)$ | **0.500** | 62.876 | 0.191 |
| $(6, 0.1)$ | **0.456** | 5.718 | $-0.027$ |
| $(4, 0.03)$ | **0.456** | 10.810 | 0.097 |

Table 12: Clustering coefficients of WS networks

| WS, $(d, \delta)$ | $\gamma_c$ | $L_p$ | Ratio |
|---|---|---|---|
| $(6, .001)$ | 0.599 | **21.188** | 0.209 |
| $(4, .0075)$ | 0.489 | **21.264** | 0.182 |
| $(6, .005)$ | 0.592 | **14.310** | 0.211 |
| $(4, .015)$ | 0.473 | **14.253** | 0.174 |
| $(6, .009)$ | 0.585 | **12.081** | 0.137 |
| $(4, .0225)$ | 0.467 | **12.171** | 0.125 |

Table 13: Clustering coefficients of WS networks

$\mathrm{SBM}(N, M, p_1, p_2)$. Note that the (expected) number of edges, denoted by $|\mathcal{E}|$, is

$$|\mathcal{E}| = \frac{p_1}{2} N(\frac{N}{M} - 1) + \frac{p_2}{2} \frac{N^2}{M}(M - 1). \tag{68}$$

Now, we fix $|\mathcal{E}|$, and choose the pair $(p_1, p_2)$ under a fixed $|\mathcal{E}|$ in (68). The aim of fixing $|\mathcal{E}|$ is to guarantee that the transmission of the disease would not be affected by edges.

**V-SBM** Now, we consider a variant of SBM, denoted by V-SBM. Different from SBM, we only allow nodes in cluster $i$ to connect to nodes in successive clusters (the neighbor clusters). Denote the V-SBM as V-SBM$(N, M, p_1, p_2)$. Similarly, the expected number of edges, denoted by $|\mathcal{E}|$, is

$$|\mathcal{E}| = \frac{p_1}{2} N(\frac{N}{M} - 1) + p_2 \frac{N^2}{M}. \tag{69}$$

Now, we fix $|\mathcal{E}|$, and choose the pair $(p_1, p_2)$ under a fixed $|\mathcal{E}|$ in (69). The aim of fixing $|\mathcal{E}|$ is to guarantee that the transmission of the disease would not be affected by edges.

## M.2 The impact of $\gamma_c$ and $L_p$ individually

In this subsection, we investigate the role of $\gamma_c$ and $L_p$ individually, not through the common factor $\delta$. We consider different WS networks with degrees $d = 4, 6$, and then adjust the rewiring probability $\delta$, such that one of $(\gamma_c, L_p)$ is almost constant, and the other is varying. We can see that the trend is similar to what we observed by varying $\delta$ in Table 6.

