# OpenReview forum: "Containing a spread through sequential learning: to exploit or to explore?"
_TMLR — Accepted by TMLR_

### Review · Reviewer_WZi5 · 2022-12-28

**Summary Of Contributions:**

The authors study interventions to prevent the spread of a process through a network, motivated by contact tracing of infectious disease. They consider a setting where the network structure is known, but the individual state of nodes, which follow a Susceptible Latent Infectious Recovered discrete time compartmental model, is unknown. The decision maker has a budget of tests per time period. The decision-maker must decide which nodes to test at each time step. If a node tests positive, they are isolated until they are recovered.

The authors show that, under the simplifying assumption of myopic (one-step) minimization of infection, the optimization problem is supermodular, increasing and monotone. The resulting optimization is still expensive for large graphs. They consider further simplifications, deriving an upper bound for the objective and estimating the node state distribution using past test results. This is, however, a pure exploitation policy. The authors add an exploration component and prove that this results in an improvement of a constant factory in a line network.

The last methodological contribution is a message-passing framework that is used to estimate the state of nodes based on test observations in a computationally efficient way and will provable converge to the true probabilities.

In simulations, they compare their new methods with baselines, such as random testing, contact tracing subject to a test budget and a logistic regression-based predictor of infection on both synthetic and and a real data network (Copenhagen Networks Study). Their policies appear to outperform baselines.

**Audience:**

Yes

**Broader Impact Concerns:**

I do not have broader impact concerns for this paper. (I disagree that a broader impact statement is needed for this paper.)

**Claims And Evidence:**

Yes

**Requested Changes:**

I do not think it is accurate to say that contact tracing depends on knowledge of the network topology. I believe that the “practical” contact tracing methods cited depend only on the neighbors of cases that are to be traced, not that the _entire_ network topology (certainly Kucharski et al. is in that category). They are not “network-aware” in some sense. This is in contrast to the authors proposed policies, which require the entire network (see weakness #3).

There are some papers that consider sequential testing and isolation policies. In addition, the authors make a claim that there are "only two relevant papers". It depends exactly what you consider relevant, but other papers the authors could consider.
- Meirom et al., ICML 2021. https://icml.cc/virtual/2021/spotlight/9910. This seems closest to the authors' setting, and I believe could serve as a direct comparison. Unfortunately, there is no publicly available code, as far as I know, and the published experiments don't look comparable.
- Kucharski et al., Lancet Infectious Diseases 2020 does consider what happens if you test n% of the population every day.
- Perrault et al., https://www.medrxiv.org/content/10.1101/2020.11.16.20227389v2
- Also potentially related is work by Ou et al. in active case finding: https://ink.library.smu.edu.sg/cgi/viewcontent.cgi?article=6148&context=sis_research. (see below)

Definitions:
- Contact tracing may trace contacts either forward in time from an index case (forward contact tracing) or backward in time (i.e., finding the individuals who were contacts of the index case in the period before they developed symptoms, e.g., https://www.nature.com/articles/s41567-021-01187-2). It is unclear whether the authors’ statements in the intro are referring to both forward and backward contact tracing or just forward.
- The authors identify shortcoming of contact tracing:
“(i) They are not able to prioritize nodes based on their likelihood of being infected (beyond the coarse notion of contact or lack thereof)
(ii) they do not incorporate any type of exploration”
It is not clear to me that (i) is a shortcoming of contact tracing—it might be a shortcoming of particular contact tracing policies, but it’s not an inherent shortcoming of contact tracing.
(ii) is a shortcoming of contact tracing in the sense that it would fall under the term “active case finding” instead. See, e.g., thelancet.com/journals/lanpub/article/PIIS2468-2667(21)00033-5/fulltext

Domains would be helpful in the notation table, e.g., \sigma \in {I, S, L, R}

Benchmark policies “Logistics regression” -> "Logistic regression"

It is advisable to remove grant numbers from Acknowledgments to protect the authors' anonymity.

**Strengths And Weaknesses:**

Strengths:
- To my knowledge, this particular model has not been studied before and the supermodular formulation and the message-passing algorithm are novel.
- I liked the discussion of exploration and exploitation. It is cool that you can explicitly "see" the impact of exploration.
- I found the paper to be well-written overall. Clear about contributions, with a clear mathematical exposition—enough detail so that the argument can be followed in the main body and lots of detail in the appendix.

Weaknesses:
- I have some complaints with the intro, both in terms of definitions and related work, see requested changes. I believe that the related work could be improved (and the authors make some unsupported claims here), but the authors' central contributions are supported.
- There are other baselines that could be compared, particularly those that use reinforcement learning, see requested changes. I'm not sure that is fair to ask the authors to do all these comparisons, but I believe that they would likely outperform the authors' methods.
- There is a central simplifying assumption being made that the entire contact network is acquired at each step. I think it is OK for the authors to make this simplification, but it is important to recognize that there is no real-world setting (that I know of) where the entire contact network is available. Usually, specific queries have to be made to get existence of any edges.
- The paper is centered around COVID-19 in a way that is somewhat unsatisfying, given that the applications of this kind of method are likely to be around other infectious diseases. (The mutations of SARS-CoV-2 mean that the spread is very hard to stop in the way the authors propose.)
- Symptoms do not appear anywhere. For some infectious diseases, symptoms are a very reliable signal of infectiousness (e.g., Ebola). Many contact tracing policies use symptoms in some central way.

Post-response update:
I understand the criticisms of the other authors, but I still think there is enough to justify acceptance.

The authors gave a thorough response that fixes substantial issues with the framing (the original version suffered from overclaiming in several areas). I am OK with the fact that there is not a fully precise theoretical advance here.

---

### Review · Reviewer_PytJ · 2023-01-04

**Summary Of Contributions:**

The paper studies "testing and controlling" a disease (or any other harmful entity) spreading in a network.  The authors consider an indepdent spread model similar to "independent cascade", where each infected node independently attempts to infect its neighbors with a certain probability.  At each time, the algorithm may choose a given number of nodes to test.  If a node tests positive, then that node is immediately isolated and stops infecting its neighbors.  The goal is to minimize the cumulative number of infected nodes.

The authors present a family of heuristic methods (guided by theoretical insights) for the problem.  The authors observe that at each time, the number of newly infected nodes is supermodular in the set tested at the current time.  Based on this observation, the authors propose an algorithm which greedily chooses the set of nodes to test based on the marginal gain of each node.  This gives an approximation guarantee for each time step, but not over the entire process (which is why the algorithm is heuristic).  The authors further give an relaxation of the greedy algorithm, where an upper bound (the "reward") of the marginal gain of each node is computed before hand, and the ones with the highest rewards are tested.  The authors call this "exploitation".  Taking one more step forward, the authors propose an algorithm where each node is tested with probability proportional to the above reward.  The authors call this "exploration".  When implementing these algorithms, the authors propose a heuristic method ("message-passing") for estimating the posterior belief of the states of all nodes at each time.  The estimation method disregards any correlation between nodes.

The authors then test their algorithms on synthetic and real networks.  The results suggest that "exploitation" and "exploration" tend to outperform the 3 baseline methods.  The authors also vary parameters of the environment and study how that affects the performance of different algorithms.  The results roughly align with intuition.

**Audience:**

Yes

**Broader Impact Concerns:**

I don't have concerns.

**Claims And Evidence:**

No

**Requested Changes:**

(Also including detailed comments here.  The authors can use their judgment regarding the more subjective suggestions.)

Sec 1, second paragraph: "... such SIR models ..."

Last paragraph on page 2, "... knowledge vs. exploration ..." (and later "exploration vs. explotation ..."): Extra space after "vs." (consider using, e.g., "vs.{}").

"Covid-19" vs "COVID-19"

Eq (2), problem formulation: So the budget constraint is in-expectation.  Any justification for this?  Do you know anything (e.g., hardness results) when the constraint must hold with probability 1?

Relatedly: What does the algorithm know about how the network evolves?  I guess complete knowledge is a bit impractical, but you have to know something?  Overall I feel there's not enough information in the modeling section, and many important details of the model are only introduced later when they are referred to (see comments below).

Eq (5): So you assume infections happen independently even restricted to a single node.  This is fine (cf. the independent cascade model in influence maximization) but it might be helpful to also explain in words.

Lem 1: This should be trivial given all relevant details of the model.  In particular it looks like you assume any node which tested positive at time t is immediately isolated, which I would put in the modeling section (since it's not something that you prove).  What about nodes that tested positive before t?  Are they removed from the network?  I'd also discuss this more formally in the modeling section.

Algo 1: So this has some approximation guarantee for the single-stage problem, but can you say anything about how good it is for the actual objective in Eq (2)?  I wouldn't say it's "near-optimal" without such a guarantee.

Algo 3: Well I guess this is where you need the budget constraint to be in-expectation...  Still I feel a hard constraint would be more natural.

Eq (13): It looks like you are assuming quite strong independence over nodes, which in principle is incompatible with the belief updating process.  I'd remark somewhere that this is just a heuristic estimation.  (Remark 2 sort of touches on this but that remark is more about the update caused by observations at time t, and that's not the only source of dependency.  In particular, spreading also introduces dependency.)

Algo 3: I wouldn't call this "exploration" since (as the authors also say in the paper) the algorithm sort of balances between exploration and exploitation, with respect to estimated rewards.

Fig 5: What is Greedy?  Algo 1 I guess?  Also, why is Greedy not in Fig 6?

**Strengths And Weaknesses:**

Strengths:

The model is natural and practically relevant.  The algorithms and the message-passing subroutine are simple and intuitive (which is good).  The authors provide justifications for their approach by analyzing several stylized networks.  The experimental results look promising and align with intuition.


Weaknesses:

None of the algorithms have a real theoretical guarantee.  The model is never fully defined, and the mathematical reasoning sometimes depends on implicit assumptions that are either vaguely mentioned or never mentioned (see detailed comments).  In terms of empirical results, I'd like to see the algorithm tested on different models of synthetic networks and other real networks.


Justification of evaluation:

I don't think "whether the claims are supported by evidence" always has a clear binary answer.  Here I'm interpreting it in a strict sense since it's the main evaluation criterion of TMLR.  Given the weaknesses mentioned above, I'm reluctant to say that all the claims (in particular, those in the abstract) are supported by accurate, clear and convincing evidence.  In particular I'd tone down any claim that involves "theoretical guarantees" or things like that.

---

### Review · Reviewer_9us1 · 2023-01-07

**Summary Of Contributions:**

The paper uses a discrete-time compartmental model with type space S-L-I-R (Susceptible-Latent-Infected-Recovered) to model the spread of an epidemic (e.g., Covid-19) over a graph. The objective is to come up with a testing policy to minimize the expected number of infected nodes after $T$ rounds of propagation, subject to testing budget constraints at each $0 \leqslant t \leqslant T-1$. A node that tests positive is isolated and pruned from the graph. Instead of studying the global optimization problem over T periods, the authors instead focus on solving single period problems myopically where the objective is to minimize the number of new infections spawned in the next period. The rest of the paper is focused on the analysis of and approaches to the single period problem. Numerical experiments are provided at the end.

**Audience:**

Yes

**Broader Impact Concerns:**

Since the paper is anchored around Covid-19, I believe an impact statement is warranted. The authors ought to clearly state the limitations of their work, the assumptions that go into it, and potential risks and pitfalls of their algorithms in a separate section.

**Claims And Evidence:**

No

**Requested Changes:**

1. Figure 1 should be S-L-I-R.
2. That the states of the nodes are independent is a very critical assumption that underlies everything else. In my opinion, this warrants a detailed discussion -- something that is currently lacking.
3. On page 8, where is "OPT" defined? I understand from the context what this might be, but please define it formally.
4. Please state the performance guarantee of Algorithm 1 formally. Clearly state what constant factor approximation is achieved by this algorithm.
4. Footnote 3 should be $x\in\mathcal{X}\backslash \mathcal{B}$ (please correct the typo)
5. As far as I understand, the set returned by Algorithm 1 should be $\mathcal{K}^\pi(t) = \mathcal{A}_i$ (not its complement). Please fix this.
6. In Algorithm 2, $\underline{w}(0)$'s and $\hat{r}_i(t)$'s haven't appeared before. Please state and describe everything clearly.
7. How is $\hat{r}_i(t)$ computed? I could not find this anywhere.
8. Step 2 of Algorithm 2: K(t) should be B(t).
9. Where are the performance guarantees of Algorithms 2 and 3?
10. Typo in Algorithm 4 first line: $\in$ is missing.
11. Theorem 2: What does with probability approximately 1/e mean? What is the norm? What is the mode of convergence?
12. Theorem 3: The result basically intends to convey that by injecting a small amount of randomness (exploration) in the testing policy, one can reduce cumulative infections by a constant factor. But this needs to be stated in a "formal" language.
13. In Section 7 (Conclusions and Future Work), first paragraph, the authors claim that they propose a "near-optimal" testing policy. This is not true. Their policy (Algorithm 1) is based on reference [52] and only provides a constant factor approximation to a supermodular minimization problem (known to be NP-hard). What am I missing here? Is the approximation ratio best possible? This line appears to be an unsubstantiated claim.

Unfortunately, this paper, in my opinion, is below the bar in its current form, and needs substantial work.


**Strengths And Weaknesses:**

The paper is thematically relevant to the times, and studies an important problem. It also proposes some interesting first-order approaches. In particular, I like the Thompson Sampling-esque idea used in Algorithm 3 to probabilistically test nodes that are most likely to be infected.

My grouse is that the paper is not written in a formal manner and lacks technical results (at least in the main text). The paper (to my mind) is positioned as a theory work, yet lacks any formal statements of performance guarantees, etc. I have listed my major concerns, errors and typos spotted in the next section.

---

### Decision · Action_Editors · 2023-02-22

**Recommendation:** Accept as is

**Comment:**

This paper studies the problem of stopping the spread of a disease in a network. The key idea is to greedily minimize the number of newly infected nodes in the next round by testing and isolating the nodes in the current round. The authors propose two notable variants of their main algorithm, which optimize a more tractable objective and estimate the unknown distribution of the states of the nodes. The proposed algorithms are evaluated empirically.

This work is on a good topic, which became highly relevant recently due to COVID. The original paper suffered from several issues, such as over-claimed contributions and imprecise writing given that this paper tries to make theory contributions. These were addressed in the updated paper. I agree with the reviewers that the theory contribution of this paper is limited. On the other hand, it is a hard problem and the algorithmic contributions partially make up for it.

**Audience:**

Yes. Spread of a disease in a network is analogous to influence spread in a social network. The latter problem is very popular, and appeals to both theoreticians and practitioners of ML.

**Claims And Evidence:**

Yes. The main contribution are theory-motivated algorithms for stopping the spread of a disease in a network. The usefulness of the algorithms is demonstrated empirically.